

# Impact of cirrus on extratropical tropopause structure

Nicolas Emig [1], Annette K. Miltenberger [1], Peter M. Hoor [1], and Andreas Petzold [2,3]

[1]Institute for Atmospheric Physics, Johannes Gutenberg University Mainz, Johann-Joachim-Becher-Weg 21, 55128 Mainz, Germany
[2]Institute of Climate and Energy Systems 3 – Troposphere, Forschungszentrum Jülich GmbH, Jülich, Germany
[3]Institute for Atmospheric and Environmental Research, University of Wuppertal, Wuppertal, Germany

**Correspondence:** Nicolas Emig (niemig@uni-mainz.de)

**Abstract.** Diabatic processes are essential in shaping the thermodynamic and chemical structure of the extra-tropical transition layer (ExTL). Cirrus may play a vital role due to associated latent heating and their influence on radiative and turbulent properties. Here we present for the first time in situ observations of the ExTL thermodynamic structure in- and outside cirrus by utilizing a dual-platform approach. The observational data were collected during the AIRTOSS-ICE campaign. Earlier analysis by Müller et al. (2015) suggests that the observed cirrus had formed in stratospherically influenced air masses based on measured $N_2O$ mixing ratios. The dual-platform approach reveals substantial disturbances in the vertical profile of potential temperature with a weakened stratification inside the cirrus and sharpening above.

Lagrangian analysis based on high-resolution ICON simulations suggests that cirrus related radiative cooling and latent heating are instrumental in the formation of the observed disturbed potential temperature profile. Radiative cooling and to a lesser degree turbulent heat and momentum transport result in substantial PV production in the upper part of the cirrus and a steepening of the vertical potential vorticity gradient. The simulation reproduces key aspects of the in situ observations and the larger-scale evolution as evident from satellite and radiosonde data. Our analysis further indicates that the cirrus particles formed in an already moist ExTL air mass over Southern Germany about 12 hours before it being sampled over the North Sea.

Our findings underline the importance of diabatic cloud processes for the thermodynamic structure of the ExTL and potential cross tropopause exchange.

## 1 Introduction

The upper troposphere/lower stratosphere (UTLS) region plays an important role for the radiation budget of the atmosphere. Riese et al. (2012) and Forster and Shine (2002) showed the high sensitivity of this region to changes in composition, especially in ozone and water vapor, which is affected by irreversible exchange processes between the troposphere and the stratosphere. Subject to these exchange processes is the formation of the extratropical transition layer (ExTL, WMO (World Meteorological Organization) (2002)) or 'mixing layer' (Danielsen, 1968; Hoor et al., 2002; Fischer et al., 2000). The ExTL constitutes a layer above the extratropical dynamical tropopause (typically identified as the 2 pvu isoline of potential vorticity), i.e. around the thermal tropopause, where the composition is directly influenced by the extratropical troposphere (Hoor et al., 2004; Pan et al., 2004; Hegglin et al., 2009; Zahn et al., 2014; Petzold et al., 2020). It is affected by transient and frequent mixing events leading





to varying strengths of gradients in composition, connecting typical tropospheric and stratospheric values of various trace gases. Transport into this layer occurs mainly through three different pathways (e.g Holton et al., 1995; Gettelman et al., 2011) with seasonally varying contributions (Hoor et al., 2005; Bönisch et al., 2009): (i) Quasi-isentropic exchange from the tropical upper troposphere across the subtropical jets (e.g. Ray et al., 1999; Haynes and Shuckburgh, 2000), (ii) diabatic downward transport from the stratospheric overworld as part of the global stratospheric overturning circulation (Brewer-Dobson-circulation, BDC,

Butchart (2014)) and (iii) stratosphere-troposphere exchange (STE) across the extratropical tropopause (e.g. Sprenger and Wernli, 2003; Škerlak et al., 2014; Stohl et al., 2003; Engel et al., 2006). STE can be diagnosed by analysis of the chemical composition. Various trace gases with sources in either the troposphere or the stratosphere such as e.g. $N_2O, H_2O, CO$ or $O_3$ exhibit sharp vertical gradients in the ExTL as do aerosol or cirrus particles and therefore varying strenght of the gradients potentially indicates STE (Joppe et al., 2024; Pan and Munchak, 2011). Notably, isentropic composition gradients indicate the

effect of irreversible transport and subsequent mixing (Kunkel et al., 2019; Lachnitt et al., 2023).

Since potential temperature as well as potential vorticity are conserved quantities under adiabatic conditions the processes involved in the formation of the ExTL have to be of diabatic nature, i.e. are related to turbulence, latent heating, or radiative processes. Turbulence at the dynamical tropopause can result in STE due to the associated irreversible mixing. Waves on planetary and synoptic scales, as well as the breaking of gravity waves, can lead to horizontal and vertical wind shear,

thereby favoring the occurrence of turbulence in the tropopause region (Kunkel et al., 2019; Kaluza et al., 2021). Convection also may generate gravity wave induced shear, turbulence and mixing across the tropopause (e.g. Mullendore et al., 2005; Homeyer, 2015; Homeyer et al., 2017) and thereby transport of large amounts of water into the UTLS. Latent heating due to phase changes of water during cloud formation alters local temperatures, thermal stratification, and potential vorticity (PV). Radiative processes likewise alter the local temperature and in regions with vertical gradients in optical properties of the at-

mosphere, e.g. through gradients in chemical composition or at cloud boundaries (Forster and Shine, 2002), they impact the thermal stratification and PV. The alteration of thermal stratification and PV can result in the transfer of air between stratosphere and troposphere or vice-versa. Furthermore, local temperature changes through latent or radiative heating can foster buoyancy driven turbulence (e.g. Spichtinger, 2014) and promote mixing. The occurrence of turbulence and cirrus clouds in the region of the dynamical tropopause may therefore be key to quantifying STE (Spreitzer et al., 2019) and to understanding

the spatio-temporal variability of ExTL properties.

The dynamical environment and microphysical properties of cirrus clouds impact cirrus latent and radiative heating profiles and therefore their role in STE and the ExTL structure. In particular the microphysical properties of cirrus clouds have been suggested to depend on the cirrus formation mechanism, which can be divided into (at least) two categories (Krämer et al., 2016): (i) Liquid-origin cirrus forming by freezing of liquid cloud droplets transported to cold temperatures from the lower,

much warmer troposphere. This type of cirrus tends to form in updrafts extending over a large fraction of the tropospheric depth (e.g. warm conveyor belts (Wernli et al., 2016) or deep convection). In contrast, (ii) in situ cirrus forms by ice nucleation at temperatures below 235 K without the air becoming saturated with respect to liquid water. Hence, in situ cirrus is typically encountered at very cold temperatures and slow to moderate updrafts. Thus in situ cirrus is more likely to be found in close proximity to the dynamical tropopause than liquid origin cirrus. The properties of in situ cirrus can further vary according





to updraft speed (Krämer et al., 2016): Slow updrafts associated with frontal systems lead to low ice water content and long lifetimes while fast updraft causes higher ice water content and shorter lifetime.

There have been few observations of ice particles above the dynamical or thermal tropopause (e.g. Keckhut et al., 2005; Pan and Munchak, 2011; Müller et al., 2015; Rolf et al., 2012). Cirrus particles in the ExTL could be the result of extratropical stratosphere-troposphere exchange. However, they could also form from previous water vapour injections and subsequent adi-

abatic cooling by further uplift or advection to colder regions. Long range transport from low latitudes is highly unlikely since temperatures at the tropical tropopause layer (TTL) are lower than in the midlatitudes and saturated air passing the TTL would not be saturated anymore once it reaches the midlatitudes (Newell, 1982; Dessler et al., 1995). Satellite-based observations of ExTL cirrus have been made (Zou et al., 2020), for example with Lidar (CALIOP (Avery et al., 2012)) or limb sounding measurements (MIPAS (Fischer et al., 2008), CRISTA (Spang et al., 2015)). Despite the fast spatio-temporal coverage of the

observations, an essential disadvantage of these remote sensing methods is the low vertical resolution and the reliance on reanalysis models to determine the tropopause height, which makes it difficult to analyze STE associated with small-scale processes.

In contrast, Müller et al. (2015) presented a case showing extratropical cirrus occurrence in (chemically) stratospheric air utilizing simultaneous aircraft-based in situ measurements of ice particles and $N_2O$ mixing ratios. The measurements were taken

during the AIRTOSS-ICE campaign in 2013 over Northern Germany. For the localization of the observed ice particles in the ExTL, Müller et al. (2015) use $N_2O$-measurements to determine a chemical tropopause. They argue that negative deviations from the tropospheric background mixing ratios in $N_2O$ can only be explained by irreversible mixing with stratospheric air. On one occasion, measurements of ice particles coincided with $N_2O$ levels below the tropospheric background values. Backward air mass trajectories calculated on the basis of ECMWF operational analysis data suggest an uplift across the 2 pvu isosurface

in the 3 h before the measurements. The Lagrangian analysis further indicated, that the ice particle formation occurred during slow ascent in the upper troposphere and subsequent transport into the lower stratosphere.

In this study we extend the analysis of Müller et al. (2015) by newly available Lagrangian diagnostics with much higher temporal and especially vertical resolution based on ICON simulations (Miltenberger et al., 2020). The simulations include diabatic tendencies to gain insight into the formation of the observed stratospheric cirrus as well as its impact on the tropopause and

ExTL structure.

Corresponding to the higher model resolution we extended the measurement data presented in Müller et al. (2015) by simultaneous measurement data from a second platform (the TOSS, (Frey et al., 2009; Klingebiel et al., 2017)). These additional data allow for the calculation of vertical gradients of temperature and potential temperature, i.e. static stability.

The model based hypothesis about cirrus formation and cirrus impact on ExTL structure are supported by comparison with

the in situ measurements as well as additional observational data, i.e. satellite observations of cirrus evolution and radiosonde measurements of the thermodynamic structure at upstream locations.

In particular, the research presented in this study addresses the following questions:

1. What is the thermodynamic structure associated with ExTL cirrus occurrence?



2. Where and how do the observed stratospheric cirrus particles form?

3. Does the cirrus cloud modify the local ExTL (thermo-)dynamical structure?

The paper is structured as follows: We will first give a short overview about the observations and model tools in section 2. Section 3 contains a short overview of the synoptic situation, presents observations of the thermodynamic structure and the cirrus occurrence with respect to the tropopause on the basis of the simultaneous observations from the TOSS and the aircraft, and compares the ICON model thermodynamic structure in the measurement region to the in situ data. In section 4 the model based Lagrangian history of the measured air mass, the cirrus formation mechanism and the cirrus impact on the ExTL thermodynamic structure are discussed. Section 5 presents satellite observations and radiosonde data in locations upstream of the measurement region to provide evidence for the plausibility of the modelled air mass history. Finally, we close with a summary and brief discussion of our main findings in section 6.

## 2 Methods

### 2.1 In situ measurements

Measurements were carried out during the AIRTOSS-ICE campaign with a Learjet G35A of the "Gesellschaft für Flugzield-arstellung" (GFD) from Hohn, Germany. The aircraft was equipped with the University Mainz Airborne QCL-Spectrometer (UMAQS), an infrared absorption spectrometer which provided $N_2O$ and $CO$ data with a total uncertainty of 0.39 ppbv for $N_2O$ and 1.40 ppbv for $CO$ at a measurement frequency of 1 Hz. Details regarding the UMAQS-instrument can be found in Müller et al. (2015).

As a unique extension of the measurement setup the aircraft is equipped with the "AIRcraft TOwed Sensor Shuttle" (AIR-TOSS, namesake of the campaign, in the following abbreviated as TOSS). The TOSS constitutes a second measurement platform towed by the Learjet with a vertical distance of typically between 70 m and 180 m. The TOSS was released at the first level after take-off and carries additional lightweight measurement devices, which are partly redundant to the instrumentation on the Learjet and therefore allow the calculation of vertical gradients of the measured quantities as schematically depicted in Fig. 1. These redundant measurements include in particular temperatures and cloud particle number concentrations as well as size distributions. The particles at the TOSS were measured by the CCP instrument (Klingebiel et al., 2017). Particle measurements on board the Learjet were taken by the FSSP instrument. In this study both instruments will be used as cloud indicators. Temperature and humidity measurements were taken by two capacitive hygrometers on the Learjet and TOSS, respectively (ICH-sensors). These sensor types are part of the IAGOS measurement instrumental packages and have been recalibrated in 2016 (Neis et al., 2015a, b) with new estimates of accuracy and precision for temperature and humidity of $\pm 0.5$ K and 5 % $RH_{liquid}$. Potential temperature at the Learjet is calculated from the temperature measurements of the MCH sensor and the static pressure from the altimeter of the Learjet. The pressure at the TOSS was not measured directly and had to be calculated under assumption of hydrostatic equilibrium from the pressure at the Learjet and the vertical distance between the two platforms derived from GPS-data. Relative humidity was derived from the $H_2O$ mixing ratios measured by





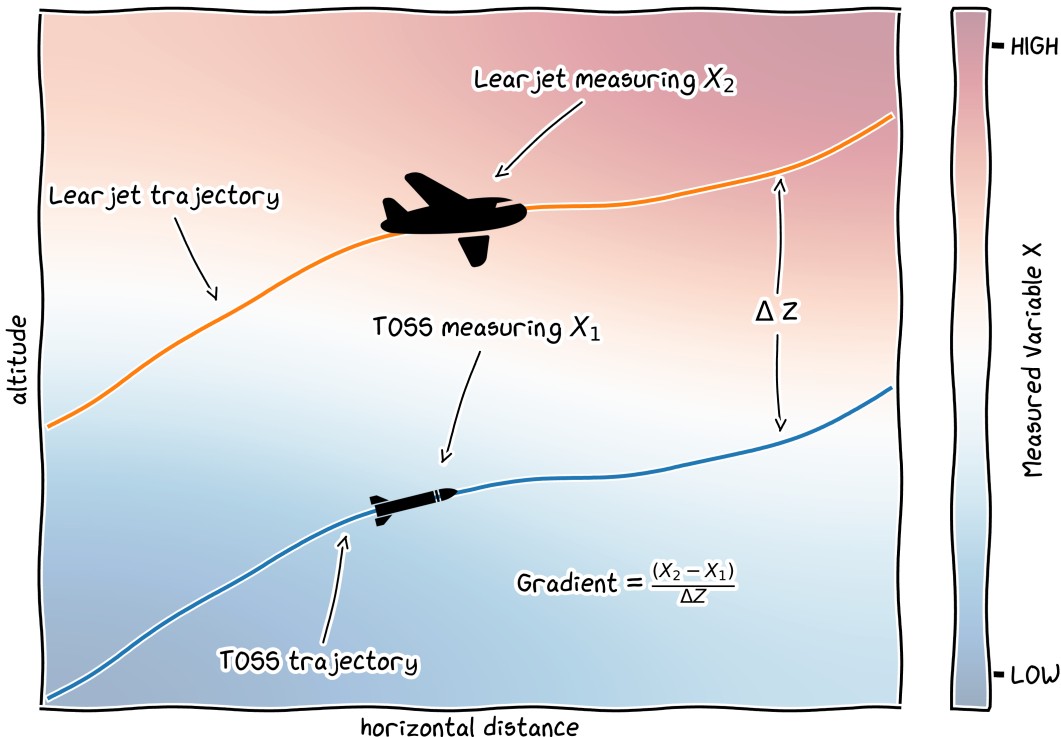

**Figure 1.** Schematic representation of the dual-platform-approach for the simultaneous measurement at two levels and calculation of vertical gradients.

the SEALDH-II instrument with an uncertainty of $\pm 1$ ppmv $H_2O$ for the condition encountered in this study (Buchholz and Ebert, 2018). During flight level changes and turns the air stream towards the TOSS inlets is perturbed and subject to turbulence. This leads to flow conditions outside the envelope of the normal flight operation and unknown effects on temperature

130 and humidity measurements. In the following we therefore only use data from horizontal flight legs with well defined flow around the sensors.

## 2.2 Model simulations

For a detailed analysis of the water vapor distribution, transport pathways and cloud formation, simulations with the numerical weather prediction model ICON version 2.6.2 have been conducted (Zängl et al., 2015). Two model simulations have been

135 conducted and are initialised from the IFS operational analysis on 12 UTC 06 May 2013 and 00 UTC 07 May 2013, respectively. Integration stops at 00 UTC 08 May 2013. A global simulation at R3B7 (effective grid spacing $\approx 13$ km) is refined with two two-way interactive nested grids over central Europe (Zangl et al., 2022). The two nests use R3B8 and R3B9 grids with effective grid spacings of $\approx 6.5$ km and $\approx 3.25$ km, respectively. In the vertical, 150 model levels between the surface and



23 km altitude are used, the spacing of which follows terrain-following smooth level vertical (SLEVE) coordinates (Leuenberger et al., 2010). This results in a vertical grid spacing of about 165 m (200 m) at 10 km (12 km) altitude. A timestep of 12 s (6 s; 3 s) is used for the integration of the model on the three grids. Sub-grid scale processes are described by the following parameterisations: Tiedtke-Bechtold convection scheme (deep convection in the global domain only, Tiedtke (1989); Bechtold et al. (2008); ECMWF (2016)), turbulence following Raschendorfer (2018), sub-grid scale orographic drag following Lott and Miller (1997), non-orographic gravity wave drag following Orr et al. (2010), cloud processes by the double-moment scheme from Seifert and Beheng (2006), and radiation by the ecRad scheme (Hogan and Bozzo, 2018).

Lagrangian analysis is facilitated by the computation of online-trajectories during the simulation (Miltenberger et al., 2020; Oertel et al., 2023). Online-trajectories are calculated based on the resolved windfield on the native grid and timestep of the ICON model in the best-resolved nest at a given geo-location.Trajectories are started at model initialisation at all grid points and vertical levels between 500 m and 15 km in three sub-regions: -2 to 5 ° E and 47 to 52 ° N, 3 to 12 ° E and 43 to 48.5 ° N, as well as 10 to 18 ° E and 46 to 50 ° N. The regions have been selected on the base of backward offline-trajectories calculated from the ICON windfields at 15 min resolution with LAGRANTO (Sprenger and Wernli, 2015). In total about 6.1 million online-trajectories have been computed. Output of (thermo-)dynamic variables as well as integrated potential vorticity (PV), potential temperature ($\theta$), and selected moisture tendencies are available every 10 min along the trajectories. For further analysis, we have selected trajectories that pass through the area of Learjet measurements (6.5 to 7.5 ° E and 54.2 to 55.2 ° N, 8 to 12 km altitude) between 14 UTC and 16 UTC 07 May 2013.

## 3  UTLS structure in the measurement region

### 3.1  Synoptic situation

The synoptic situation on the day before the flight, i.e. 06 May 2013, is characterized by southerly flow over Germany associated with a ridge and weak trough at its western flank. Until the time of measurement the ridge shifted slightly towards the east, but southerly flow still prevailed over Germany. The trough developed to a filament accompanied by streamers of stratospheric air and filaments of dry air between an approaching weak cyclonic system from the west and the eastward shifted ridge. An associated elongated PV-filament extended southwards to central Italy. In connection with the southerly flow at the western flank of the ridge high humidity is transported to the north, just adjacent to the dry filament leading to strong humidity gradients in the upper troposphere over western Germany. Satellite images suggest that optically thin cirrus clouds form in the the humid southeasterly flow at the western edge of the ridge over Southern Germany and are advected to the North Sea region (see section 5). Convective activity in the afternoon of 06 May and the morning of 07 May over Eastern Germany and Eastern Europe extending northwards from the Baltics may have contributed to the moistening of the ridge, albeit likely with no direct contribution to the ExTL cirrus discussed here (see also sec. 4). However, the optically thick upper tropospheric cirrus below the ExTL is likely partly linked to this convection (see Fig. A2). Furthermore, in the early morning of 07 May some deep clouds formed over the Benelux region likely influencing upper-tropospheric humidity westwards of the stratospheric filament.



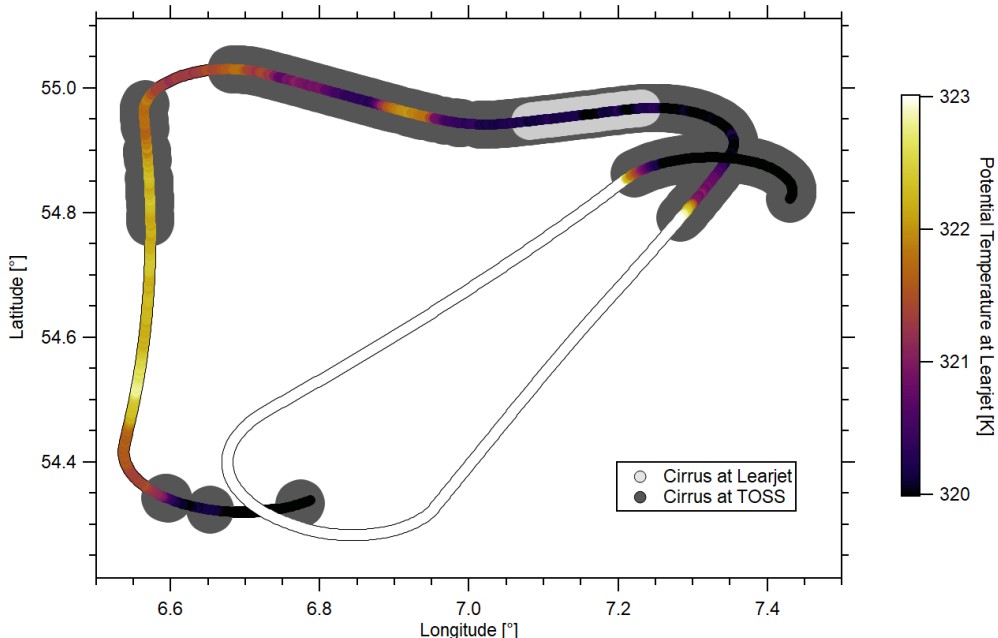

**Figure 2.** Flight path for the flight legs where stratospheric cirrus particles were encountered. Cirrus occurrence measured at Learjet (TOSS) is marked in light (dark) grey. Colors represent potential temperatures and are scaled to resolve the relevant flight leg. The following (higher) flight level is also depicted as reference.

## 3.2 Observed temperature, humidity and cloud structure of the UTLS

The section of the flight path considered in this study is depicted in Fig. 2 with occurrence of cirrus particles marked in light grey for the TOSS-platform and light grey for the Learjet. The measurements show an extended cirrus deck, with its upper edge initially localized between the two measurement platforms until the Learjet also reaches the cloud top at around 55.0 ° N

7.2 ° E. We focus on a short section from 15:08 UTC, when ice particles are first measured at the TOSS, to 15:16 UTC, before the Learjet changes altitude to the next, higher, flight leg (see Fig. 2, the section with $\theta \geq 323$ K). This section of the flight took place on a pressure level of 250 hPa and is considered to be in the stratosphere since $N_2O$ mixing ratios never reached tropospheric values of 325.9 ppbv (Fig. 3 (a), upper panel) as also discussed in Müller et al. (2015). This is consistent with PV values of 2-4 pvu indicated by ERA5 analysis at the location of the aircraft (Fig. 3 (a), second panel). The thermal tropopause,

derived from the temperature profile during the descent of the Learjet as depicted in Fig 3 (b), left panel, had an altitude of 10.35 km which is below the level of cirrus measurements at the Learjet with an altitude of 10.4 km. Note however, that the temperature profile was recorded several minutes after the cirrus occurrence. The third panel in Fig. 3 (a) shows the potential temperature measured by the Learjet (blue) and TOSS (green) along the flight segment and an indication of observed ice particles by the grey and black circles. Over the entire considered time period, cirrus particles were measured at the TOSS at a

potential temperature of $\theta =320$ K. At the beginning of this flight section the Learjet was located above the cirrus deck at a





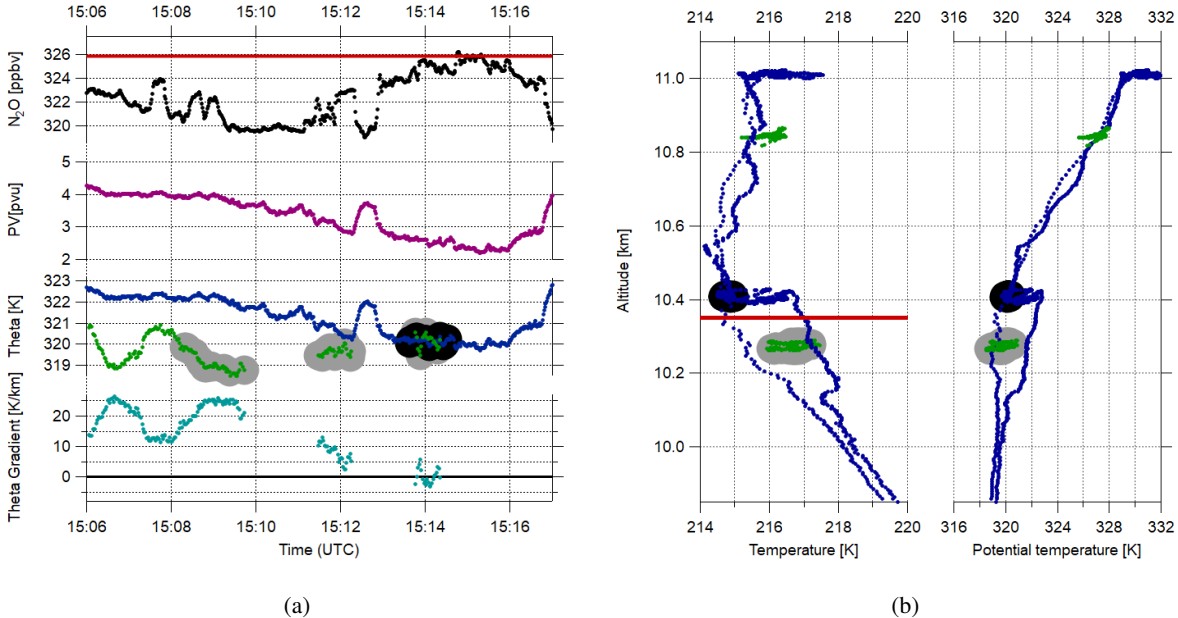

(a)                                        (b)

**Figure 3.** (a) Time series of $N_2O$ at the Learjet (black) with the chemical $N_2O$-tropopause indicated by the red line, PV at the Learjet, interpolated on the flight track from ERA5 reanalysis data (magenta), and potential temperature at Learjet (blue) and TOSS (green) and gradient of potential temperature $\Theta$ (teal). Grey (black) underlay shows the encounter of ice particles at the TOSS (Learjet). (b) Vertical profiles of temperature (left) and potential temperature (right) for Learjet (blue) and TOSS (green). Grey (black) underlay shows the encounter of ice particles at the TOSS (Learjet). The thermal tropopause, derived from the descent temperature profile, is marked in red.

potential temperature of $\theta = 322$ K. During the horizontal (i.e. isobaric) flight leg, the potential temperature at the position of the Learjet decreased slowly over time while the potential temperature at the TOSS remained constant at around $\theta = 320$ K. At the time when cirrus particles were also measured at the Learjet, the potential temperature reached $\theta = 320$ K as well.

The combined TOSS and Learjet measurements allow for the derivation of the vertical temperature gradient above and in the cirrus, which is shown in the bottom panel of Fig. 3 (a). Over the considered flight section, the vertical distance between the TOSS and the Learjet was constant. At 15:08 UTC a positive vertical gradient of potential temperature was measured, i.e. higher potential temperatures above the cirrus deck then within, as expected in the stratosphere and consistent with the vertical profile (Fig. 3 (b), right panel). However, inside the cirrus (at $\approx$ 15:14 UTC, where both platforms measured cirrus particles) the gradient of potential temperature vanishes. Hence, the measurements indicate different regimes of static stability: neutral stratification inside the cirrus and, starting at the cirrus top, high static stability above the cirrus.

Correlations of CO and $N_2O$ measurements are shown in Fig.4, colored with potential temperature (a) and relative humidity with respect to ice (b). The two species show compact mixing lines typical for the ExTL, connecting typical values of mixing





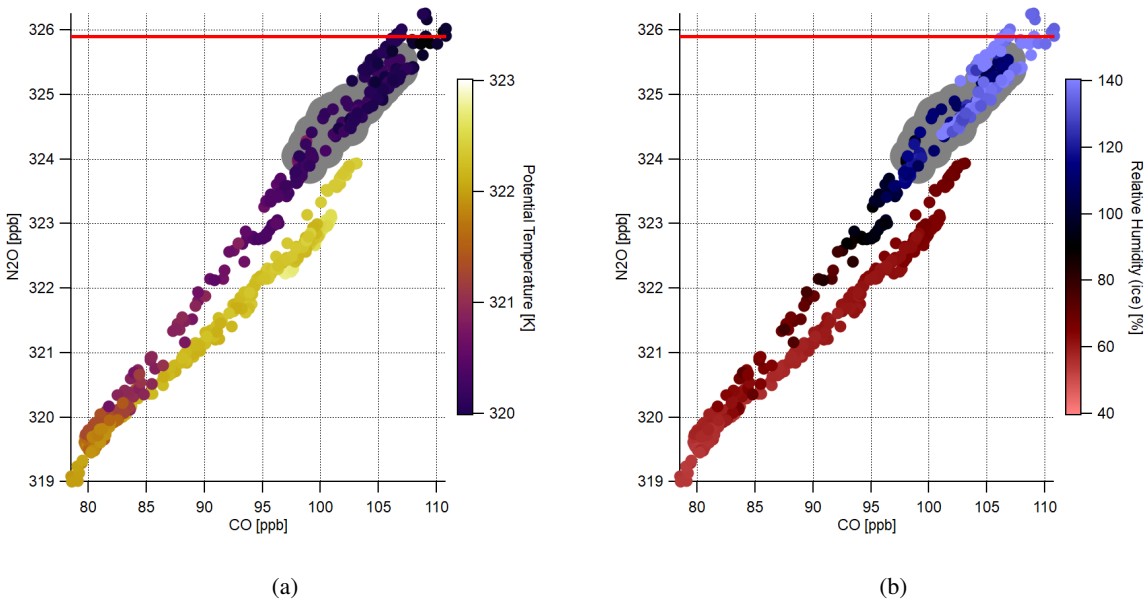

(a)                                                          (b)

**Figure 4.** Scatter-plots of $N_2O$ versus CO color coded with (a) potential temperature and (b) relative humidity with respect to ice. The $N_2O$-defined tropopause is marked as red line with lower $N_2O$ mixing ratios implying stratospheric air. Grey circles indicate cirrus occurrence.

ratios for the troposphere with such typical for a reservoir deeper in the stratosphere for both species, thereby indicating

irreversible mixing. One of the two mixing lines corresponds to the flight section at about 6.6° ° E, which is not influenced by

the cirrus at the Learjet level, albeit showing cirrus occurrence at the TOSS (see Fig.2). Values of relative humidity with respect

to ice based on measurements of $H_2O$ mixing ratios, temperature and pressure at the Learjet do not exceed 70 % in this segment

of the flight (Fig. 4 (b)). However, on the other line corresponding to the flight segment with cirrus occurrence at the Learjet

level relative humidity reaches saturation ($RH_{ice} = 100$ %) inside the cirrus and up to $RH_{ice} = 140$ % in close vicinity to the

cirrus in clear air. The highest values of relative humidity coincide with the lowest potential temperature (Fig 4 (a)) during the

entire flight leg. The presence of two distinct mixing lines suggest a different air mass history in terms of the occurrence of

mixing events and contributing air masses for air above and in the cirrus layer.

In summary, during AIRTOSS-ICE observations of a cirrus cloud located in an air mass with a distinct stratospheric chemical

signature were obtained. The direct measurement of stratification inside the cirrus and in the surroundings with the dual-

platform approach suggest reduced stability inside and strong stratification above the cirrus. The occurrence of cirrus particles

in this air mass is accompanied by high relative humidity with respect to ice.

### 3.3  Modelled temperature, humidity and cloud structure in the UTLS in the measurement region

We analyse the structure of the UTLS as represented in the ICON simulation in a 2 h window around the observations pre-

sented in sec. 3.2 and at the geographic location of the aircraft measurement, which we will refer to as "measurement area" in





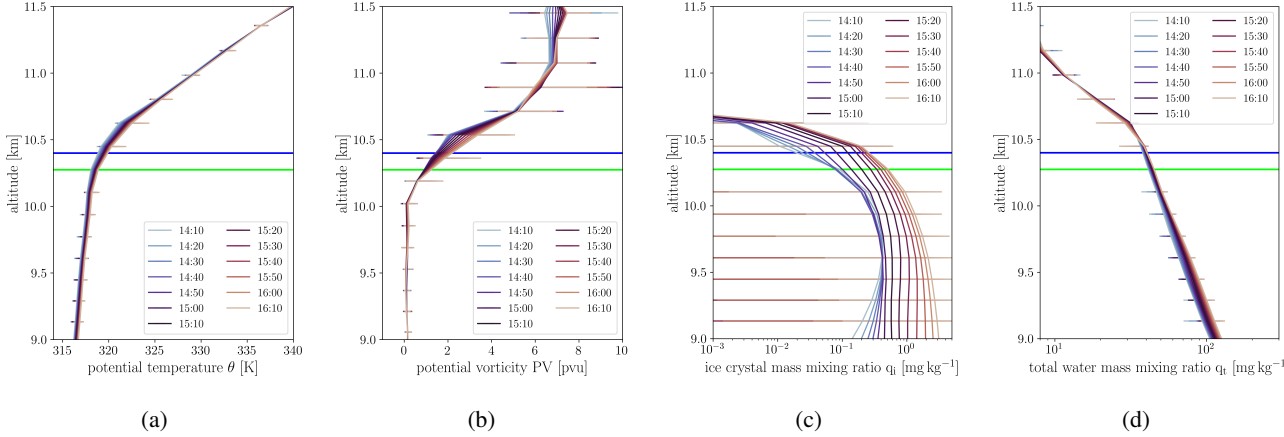

**Figure 5.** Height profiles of the average (a) potential temperature, (b) potential vorticity, (c) total water mass mixing ratio, and (d) ice mass mixing ratio from the ICON simulation. The different line colors represent profiles sampled at different times between 14:10 UTC and 16:10 UTC on 07 May 2013. Averages are taken over all grid points within $6.5 - 7.5\,°E$ and $54.3 - 55.1\,°E$; spatial variability as represented by the maximum and minimum value at a particular altitude are shown by the horizontal bars. The average altitude of the Learjet leg discussed in section 3.2 is indicated by the dark blue horizontal line and that of the TOSS by the green horizontal line. The simulation starting at 00 UTC on 07 May 2013 is shown.

the following. Exact spatio-temporal matching of ICON and aircraft is not attempted, as the finite horizontal and vertical grid-spacing as well as initial condition and model uncertainty make a perfect forecast of features on the scale discussed in sec. 3.2 extremely unlikely. Nevertheless, as we will demonstrate in the following, the model is able to capture some key features of the observed UTLS and cirrus structure. This section focuses on the UTLS structure in the simulation started at 00 UTC 07 May 2013, equivalent metrics for the simulation started 12 h earlier are shown in the Appendix. The latter simulation is also

discussed in sec. 5.

The modelled UTLS structure (potential temperature, potential vorticity, water content and ice content) and its evolution in the area targeted by the flight is shown in Fig. 5. At the altitude of the aircraft measurements (sec. 3.2) the model locates the transition region between tropospheric, low PV and weakly stratified air and stratospheric, high PV and strongly stratified air (Fig. 5 a,b). The simulated vertical profile of potential temperature (Fig. 5a) matches the transition from tropospheric to strato-

spheric stability at the aircraft location as observed (Fig. 3). PV in the two model levels closest to the aircraft altitude increases rapidly from below 2 pvu to 3 pvu on average, i.e. suggesting the location of the dynamical tropopause at around the altitude of the aircraft (Fig. 5 b). This is in slight contrast to ERA5 derived PV in Fig. 3, which places the aircraft well in the stratosphere. However, it should be noted that a vertical displacement less than the distance between two successive model levels suffices to significantly improve the consistency of ICON and ERA5. A slightly higher position of the modelled tropopause than in

observations is supported by potential temperature at the aircraft altitude being about 1.5 K colder in the model (not shown). Over the considered time period from 14 UTC to 16 UTC the vertical PV and potential temperature gradient at around 10.5 km altitude slightly increases.

The ice mass mixing ratio profiles suggest a cirrus cloud extending up to about 10.7 km in the model (Fig. 5 c). Near constant



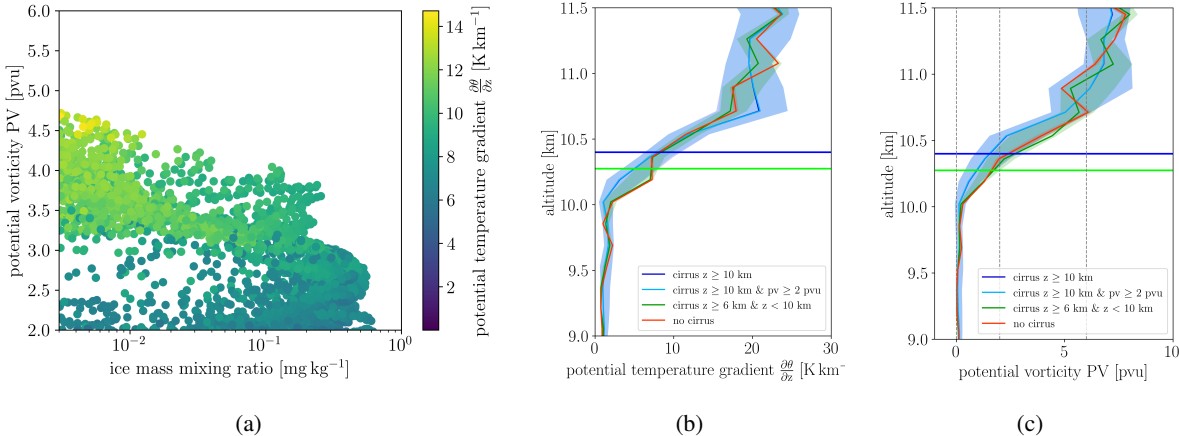

(a)                   (b)                   (c)

**Figure 6.** (a) Ice mass mixing ratio as function of the potential vorticity in the region $6.5 - 7.5\,°E$ and $54.3 - 55.1\,°E$ and times between 14:10 UTC and 16:10 UTC on 07 May 2013. The color coding indicates the potential temperature gradient at the respective gridpoints. Only gridpoints at altitudes above 10 km are considered. Composite profiles of (b) potential temperature gradient $\frac{\partial \theta}{\partial z}$ and (c) potential vorticity PV in the measurement area for times and location with clouds above 10 km (cyan), clouds below 10 km (green) and no clouds being present.

mass mixing ratios are simulated between 9.0 km and about 10.0 km with an increasingly rapid drop in the four model levels
above. Notably, the dynamical tropopause in the model is about 500 m lower than the cloud top suggesting the presence of
ExTL cirrus consistent with observations. Ice mass mixing ratios of around $0.1\,\mathrm{mg\,kg^{-1}}$ are found up to PV values of about
4 pvu (Fig. 6 a). Vertical profiles subsampled for model columns with and without cirrus indicate that the potential temperature
gradient in upper part of the cirrus layer is smaller than in the surrounding air and above the cirrus (Fig. 6 b). Note, however
that the modelled stratification is substantially larger than the neutral stratification observed by the Learjet-TOSS platform.
240   Consistent with the smaller vertical potential temperature gradient, PV values in the cirrus column are smaller than in the
surrounding air (Fig. 6 c). Over the analysed 2 h time window the mass mixing ratio of ice gradually increases (Fig. 5 c).

The total water content, i.e. the sum of specific humidity and cloud condensate, remains almost constant throughout the con-
sidered 2 h time period (Fig. 5 d). The humidity gradient changes rapidly at about 10.6 km altitude to larger values at higher
altitudes. Hence the tropospheric humidity gradient is continuing beyond the dynamical tropopause, consistent with the Lear-
245   jet observations and the presence of a mixing-influenced ExTL. The simulation with the earlier start time has a much higher
tropopause with less sharp gradients. Therefore the cirrus layer is found predominantly at tropospheric PV values (Fig. A1).
Hence the match with observation is not as good as in the simulation with the later initialisation date. Nonetheless, contrasting
the two simulations provides insight into the processes leading to the cirrus formation and moisture content of the lower strato-
sphere as well as potential model deficiencies in the representation of diabatic PV production, as discussed in more detail in
250   sec. 5.

In summary, the model simulation contains an ExTL cirrus expanding to well above the local dynamical tropopause as well as
reduced stratification in the cirrus. This structure of the UTLS is qualitatively consistent with the observations although there





are some small quantitative difference, e.g. concerning tropopause altitude.

## 4 Lagrangian perspective on the emergence of the modelled UTLS structure

### 4.1 Air mass origin

The UTLS structure as observed with the Learjet/TOSS framework and captured by the ICON model emerges due to advection and diabatic processes in the hours and days before arrival over the North Sea. To investigate these processes we investigate high-resolution air mass trajectories from the ICON model simulations.

The path as well as the temporal evolution of PV and ice mass mixing ratio for trajectories arriving in the measurement area is shown in Fig. 7. The different rows show different subsets of trajectories: The first two rows show trajectories passing through areas with tropospheric (557 trajectories) and stratospheric (309 trajectories) air mass composition based on the Learjet/TOSS $N_2O$ measurements, respectively. The last row shows trajectories arriving in the measurement area with PV values larger than 2 pvu and ice mass mixing ratios larger than $10^{-3} \, \mathrm{mg\,kg^{-1}}$ (1511 trajectories). The PV values in tropospheric air are predominantly smaller than 2 pvu with a few exceptions at the lowest pressures, which may be due to inconsistencies in the location of the local tropopause in the model and observations (fraction of trajectories with PV $\geq 2$ pvu $\approx 7\%$). The tropospheric air mass contains a thick cirrus layer consistent with the profiles shown in section 3.3, which forms in slowly ascending air masses over Southeastern Germany about 12 h before arrival in the measurement area (Fig. 7 a-c). The ice mass mixing ratio displays strong temporal variability likely induced by gravity waves associated with the passage over the small mountain ranges in central Germany. These wave motions are superimposed on the general slow ascent as already discussed in Müller et al. (2015). As will be discussed later, the cirrus evolution is consistent with the indication on cirrus presence from MSG satellite data (see sec. 5).

The trajectories arriving in the area with stratospheric air mass characteristics according to the observations are located on average at slightly higher altitudes, slightly further west (Fig. 7 d-f), and arrive in the measurement area predominantly in the later half of the considered time interval (not shown). They also contain an extensive cirrus cloud, but PV values are predominantly around 2 pvu. Again some spatio-temporal inconsistency between model simulation and observed situation may explain the presence of some trajectories with PV values below 2 pvu. The cirrus structure and the general characteristics of the vertical motions are not systematically different from the trajectories travelling through the area with observed tropospheric characteristics. Importantly, this also holds when considering the sub-set of trajectories with stratospheric PV values and cirrus content in the model (Fig. 7 g-i). The major differences are a shift towards even lower pressures and the dismissal of the eastern most origin region. As the general characteristics of the evolution do not differ substantially for the last two trajectory subsets, we focus in the following analysis on the trajectories with PV $\geq 2$ pvu and $q_i > 10^{-3} \, \mathrm{mg\,kg^{-1}}$. This offers a physically consistent picture on the emergence of the observed ExTL cirrus.



**Figure 7.** Path of trajectories arriving in measurement area between 13:40 UTC and 16:00 UTC with: (a-c) tropospheric marker (557 trajectories), (d-f) stratospheric marker (309 trajectories), and (g-i) $PV \geq 2$ pvu and $q_i \geq 10^{-3}$ mg kg$^{-1}$ (1511 trajectories). The left column (a, d, g) shows the geographic path of the trajectories and the color represent the pressure at which they are located. The middle and right columns show the temporal evolution of the pressure of the air parcels, the line color represents potential vorticity (b, e, h) and ice mass mixing ratio (c, f, i), respectively.

## 4.2    Cirrus cloud formation mechanism

The extensive cirrus cloud seen in Fig. 7 emerges in the 12 h before the arrival of the trajectories in the measurement area and arises by a combination of moist ExTL air over southern Germany and strong lifting of up to 1000 m (median: $\approx 500$ m) during its northward propagation. Cirrus clouds are formed in the model by deposition nucleation as indicated by the process rates from the different ice formation parameterisations in ICON traced along the trajectories (not shown). Deposition nucleation may be overestimated by the used parameterisation as it was developed for dust outbreak cases over Central Europe.





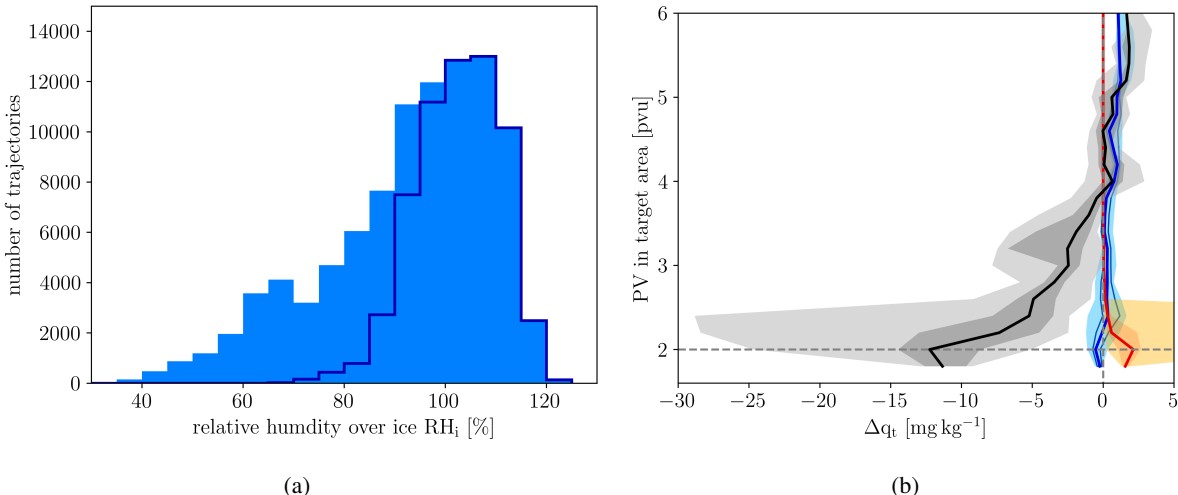

(a)        (b)

**Figure 8.** (a) Distribution of $RH_{ice}$ values in cirrus clouds forming along the trajectories arriving in the measurement area. The dark blue histograms shows the distribution, if only parcels with substantial accumulated ice nucleation rates are taken into account. (b) Change in parcel total water content $\Delta q_t$ from the start of the simulation, i.e. 00 UTC 07 May 2013, until the parcel arrives in the measurement area. The black line shows the median $\Delta q_t$; the grey shading the $25^{th} - 75^{th}$ and $5^{th} - 95^{th}$ percentile range, respectively. Similarly the change of total water from the turbulence parameterisation is shown in blueish colors and that from sedimentation into the parcels by warm colors.

Relative humidity over ice $RH_{ice}$ in the cirrus layer, which is completely below the homogeneous freezing temperature, peaks at around 120 % consistent with deposition nucleation being the dominant mode of nucleation (Fig. 8 a). If only cirrus clouds along trajectories with substantial deposition nucleation rates are considered, the small $RH_{ice}$ values are not present anymore, because air parcels acquiring ice through sedimentation are excluded (compare filled and unfilled histograms in Fig. 8 b). This distribution of relative humidity is consistent with observed relative humidity in cirrus clouds from IAGOS measurements

(Petzold et al., 2017).

  The Lagrangian diagnostics allow us to assess the change in humidity during the transport. Fig. 8 b shows the Lagrangian change of total water content from 00 UTC 07 May 2013 until the arrival of the trajectories in the measurement area. The dominant feature is a loss of total moisture of air parcels with less than 4 pvu, which coincides with the vertical extent of the cirrus deck. Hence, there is no indication that the ExTL air mass located between 2 pvu and 6 pvu gained substantial moisture

in the 12 h preceding their arrival in the measurement area. If considering the parcel's initial PV for constructing the vertical profile, the same structure emerges with small variations (not shown). The relatively high moisture content in the source region is consistent with evidence from MSG satellite data and radiosonde data over Southern Germany (as discussed in detail later, see sec. 5).





### 4.3 Cirrus impact on the PV structure and potential STE

The Lagrangian diagnostics further allow us to assess when the air parcels obtained their stratospheric characteristics at least in terms of their PV. Consistent with the visual perception of the data in Fig. 7 only few cirrus-forming trajectories passed the dynamical tropopause in the 12 h before the arrival in the measurement area (Fig. 9 a): Only 8 % of trajectories change from PV values below to such above 2 pvu in the considered time period. This is consistent with the relatively small $\Delta$PV for parcels located close to the dynamical tropopause in Fig. 9 a). However, diabatic PV modification becomes more important for parcels deeper into the ExTL: Those arriving with PV values between 3 pvu and 4 pvu gained about 1 pvu in the preceding 12 h (Fig. 9 a): Evaluation of the PV budget along trajectories indicates the strongest contribution from changes in thermal stratification by radiative processes (red line) followed by changes in thermal stratification and momentum by turbulent processes (light and dark green lines). Below 3 pvu some negative PV tendency due to latent heating from cloud microphysics is evident (blue line). Note, that the Lagrangian PV budget is difficult to close likely due to the reconstruction of PV and vorticity gradients on the staggered vertical grid followed by interpolation to the parcel positions. In particular the turbulent PV tendency field is relatively noisy on the native model grid spacing ($\Delta$x $\approx 3.25$ km, as has been noted by previous studies (Oertel et al., 2023). Composite profiles of the PV and PV modification terms along the trajectories are therefore more useful in discerning the key physical mechanisms (note: the results are qualitatively consistent with the statistics along the trajectories). The compact nature of the trajectory bundle contributing to the ExTL-cirrus and the limited vertical and horizontal shear allow a physical meaningful analysis of composite profile, which are shown in Fig. 9 b-f. Composites are constructed for different times $t$ before the arrival of trajectories in the measurement area. The PV profile indicates a narrowing of the geometric vertical extend of the ExTL, if defined as the layer between the 2 pvu and 6 pvu isoline (Fig. 9 b). The trajectories forming the cirrus cloud in the measurement area (red hatching) travel in the ExTL and the ExTL coincides with the top of the extensive cirrus deck (dots). The composite analysis suggests that the narrowing of the ExTL is mainly driven by increasing PV in the upper part of the ExTL, i.e. the highest altitude parcels. Radiative PV modification provides a consistent source of increasing PV values close to the top of the cirrus deck, consistent with the expected PV change above a cooling maximum (from longwave emission) (Fig. 9 c). Additionally, PV modification by turbulent processes is of similar amplitude, but much more localised (Fig. 9 d, f). Therefore the overall impact of turbulence on the PV of the considered trajectories is smaller than that of the radiative processes. The (partly) compensating negative PV tendency from turbulent mixing at $t = \approx -8$ to $-4$ h and positive PV tendency from turbulent mixing at $t > -2$ h contribute additionally to a reduction in the total impact of turbulence on the PV structure. In general, contributions from turbulent stratification changes are larger but of similar sign than those from turbulent momentum transport. However, the momentum transport terms are still substantial and are an important, albeit often not quantified, contribution to the PV budget. PV modification from cloud microphysical processes, i.e. latent heating and cooling, in the ExTL is generally small and decreases PV (Fig. 9 e). This is consistent with the PV modification expected above a latent heating maximum associated with the cirrus formation and ice crystal growth. The largest contributions are found around $t = -6$ h at the lower edge of the ExTL and appear to contribute to the narrowing of the ExTL by increasing PV on the tropospheric side. Note this time period is also associated with the strongest radiative PV modification and is well



aligned with the largest ice water content along the trajectories (Fig. 7 c, f, i).

340   Overall, the ICON model simulations suggest that the observed cirrus cloud formed by slant-wise lifting and gravity-wave activity (see Fig.7) in an already moist ExTL air mass originating over Southern Germany ($\approx 12$ h before arrival in the measurment area). The model provides no indication of moisture transport into the ExTL from the troposphere nor evidence for substantial STE. However, the model suggests that the extensive cirrus deck results in an enhanced PV gradient and therefore a narrowing of the ExTL in terms of its geometric depths. This results from cloud microphysical PV destruction at the bottom

345   of the ExTL as well as radiative and turbulent PV production at the top of the ExTL. Note that the modelled PV structure is consistent with the observed vertical potential temperature gradients by the Learjet/TOSS, i.e. near zero $\frac{\partial \theta}{\partial z}$ in the cirrus deck and much larger $\frac{\partial \theta}{\partial z}$ across the cloud top. Further note, that the observations suggest a much sharper cloud top than can be represented on the model vertical grid. This likely makes the discussed processes even more efficient in the real atmosphere.







(a)

(b)

(c)

(d)

(e)

(f)

**Figure 9.** (a) Change in parcel potential vorticity $\Delta$PV from the start of the simulation, i.e. 00 UTC 07 May 2013, until its arrival in the measurement area. The black line shows the median $\Delta$PV; the grey shading indicates $25^{\text{th}} - 75^{\text{th}}$ percentile range. $\Delta$PV due to temperature changes from the radiation, microphysics, turbulence, and orographic drag parameterisation are shown by the red, blue, dark green, and magenta lines (plus shading). The light green line (plus shading) shows $\Delta$PV from vertical momentum flux (turbulence and orographic drag parameterisations). (b-f) Composite profile along air parcels arriving in the measurement area with pv $\geq 2$ pvu and $q_i \geq 10^{-3}$ mg kg$^{-1}$: Variables shown are (b) PV, (c) PV tendency from radiation parameterisation, (d) PV tendency from temperature changes by the turbulence parameterisation, (e) PV tendency from microphysics parameterisation, and (f) PV tendency from momentum transport by the turbulence parameterisation. Profiles have been retrieved from 10 min Eulerian output at the native ICON grid and have been interpolated to the horizontal location of the air parcels.





## 5   Observational evidence for moisture structure in source region and modelled Lagrangian evolution of the UTLS structure

The Lagrangian analysis of the ICON simulation provides insight into the history of the observed cirrus and its role in shaping the observed UTLS temperature and moisture structure. To corroborate these model hypotheses we here use MSG satellite data and radiosonde data from Switzerland and Southern Germany, which provide observational evidence for the cloud evolution and UTLS humidity structure in the area of interest. Secondly, to further corroborate the impact of the cirrus on the UTLS thermodynamic structure from a model perspective, we discuss our second ICON simulation (initialised 12 h earlier than that discussed in section 4) that fails to reproduce the ExTL thermodynamic structure and cirrus observed be the Learjet / TOSS measurements over the North Sea (sec. 3.3 and Appendix).

Satellite data from MSG provides some information on the spatio-temporal evolution of upper-tropospheric humidity (proxy: channel at $6.2\,\mu m$), cloud extent and cloud top heights (proxy: channel at $10.8\,\mu m$), and the presence of thin cirrus clouds (proxy: difference between channels at $10.8\,\mu m$ and $8,7\,\mu m$). These images are shown for 01 UTC, 07 UTC and 13 UTC on 07 May 2013 in Fig. 10. In addition, the position of trajectories arriving between 10.5 km and 11.0 km altitude in the North Sea observation area are shown: orange contours indicate the position of trajectories calculated based on simulation started at 00 UTC 07 May 2013. In the early morning of 07 May 2013 trajectories are located along the western edge of a moist area extending from Northern Italy to the British channel (Fig. 10 a). To the west the moist region borders on a relatively dry filament. There is some indication of convection over the Netherlands (Fig. 10 b) and more tenuous cirrus further east (Fig. 10 c), which is included in the northernmost part of the trajectory source region. In the following hours the moist air mass propagates further north and consistently the trajectories propagate on the moist side of the relatively strong gradient in upper tropospheric humidity located over western Germany (Figs. 10 d, g). In this air mass the tenuous cirrus clouds present in the early morning persist and continue to coincide with the northern half of the trajectory bundle (Figs. 10 f, i). Increasingly the cirrus clouds cover the entire area, in which trajectories are found. From about 10 UTC onward optically thick clouds appear in the area covered by the trajectories, likely at slightly lower altitudes (hourly MSG images: Fig. A2). Furthermore, the time, at which the thicker cirrus appears, roughly agrees with larger ice water content at altitudes below 10 km in the ICON cross-sections along the trajectory (Fig. 9) and enhanced radiative cooling at the upper edge of the cirrus cloud. Albeit there is some indication of deep convection over Eastern Germany and the Czech Republic, the optically thick cloud over western Germany is not influenced by this directly and the optically thick cirrus rather seems to form in situ - again consistent with the ICON-based Lagrangian analysis. Hence the available satellite data confirms the model simulation in the sense, that (i) the observed cirrus formed many hours before it being sampled over the North Sea, (ii) the air mass, in which it formed was already very humid about 15 h prior to the observation, and (iii) the presence of an optically thick cirrus from about 6-8 h prior to the observation likely resulting in strong cloud top cooling.

The model analysis suggests that a key aspect for the formation of the observed ExTL cirrus is relatively high moisture content in the UTLS over Southern Germany in the early hours of 07 May 2013. Satellite analysis generally supports this idea, but for more quantitative insight we include radiosonde data from Payerne, Stuttgart, and Oberschleißheim, i.e. three




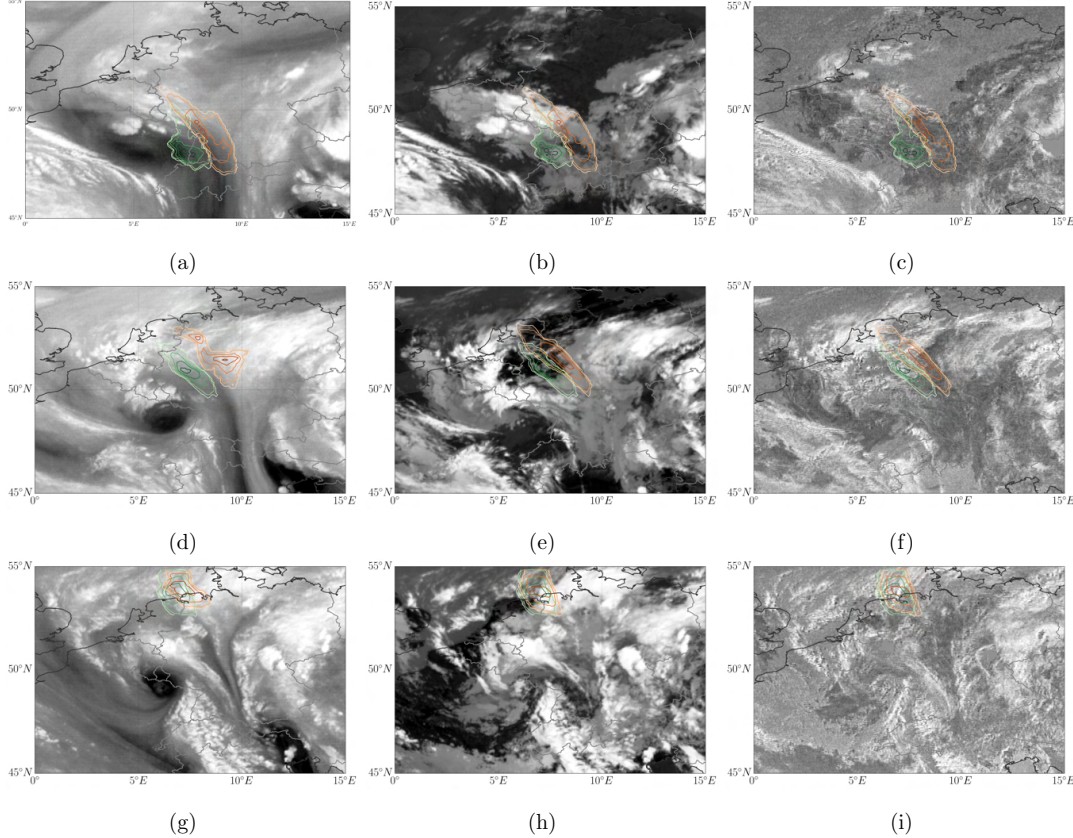

**Figure 10.** MeteoSat Second Generation (MSG) satellite images at 6.2 $\mu$m (a,d,g), 10.8 $\mu$m (b,e,h) as well as the difference signal between 10.8 $\mu$m and 8.7 $\mu$m channel (c,f,i). The rows from top to bottom correspond to observations times of about 01 UTC, 07 UTC and 13 UTC on 07 May 2013. The contours indicate the position of back trajectories from the measurement area based on the simulation initialised at 00 UTC 07 May 2013 (orange) and 12 UTC 06 May 2013 (green), respectively.

operational stations in Southern Germany and Switzerland (for location see Fig. 11 a). The temperature and humidity structure of the UTLS as observed by the operational radiosondes released at 00 UTC 07 May 2013 is shown in Figs. 11 b and A4. The

sounding data confirms a large horizontal gradient in specific humidity below and around the tropopause from west (Payerne) to east (Oberschleißheim). The Stuttgart sounding is closest to the diagnosed origin of the air mass later observed in the North Sea measurement area. The modelled specific humidity profile agrees well with the observed specific humidity data from the Stuttgart sounding, in particular with respect to the enhanced specific humidity values between 318 K and 320 K. The moist layer is not as sharply capped at its upper boundary in the model as in the observations, which likely is due to coarser vertical

grid resolution (also in the analysis data used for initialising the model simulation). The Oberschleißheim sounding suggest very humid conditions up to about 328 K, but the lapse-rate tropopause is also substantially higher at this more eastern location. The Payerne sounding shows less humid conditions in the upper troposphere, although there is some indication of an local increase



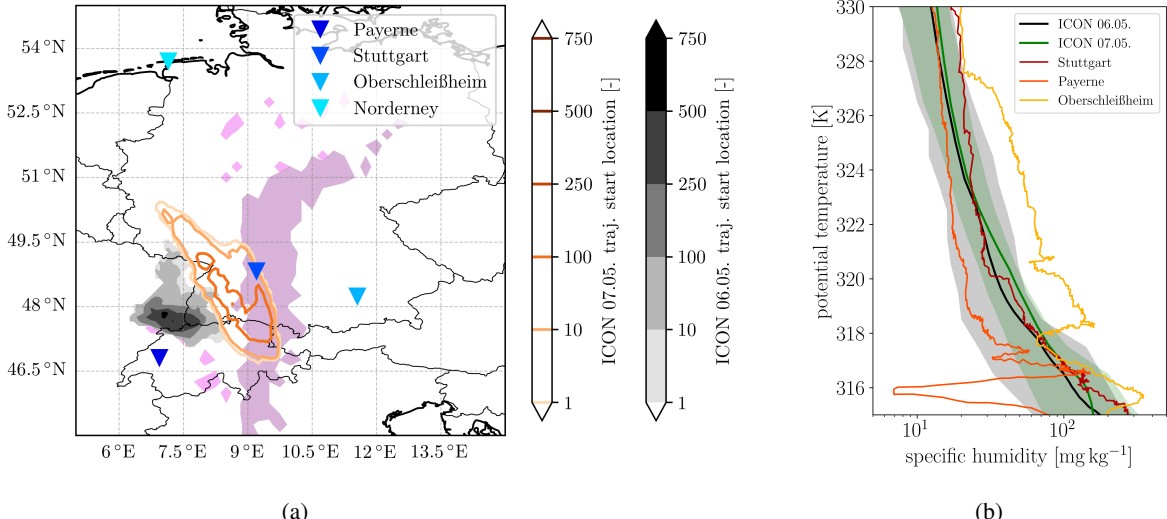

(a)                                                  (b)

**Figure 11.** (a) Location of air parcels at 00 UTC 07 May 2013. Only parcels arriving in the measurment area at altitudes between 10.5 km and 11.0 km are considered between 14 UTC and 16 UTC on 07 May 2013. The grey shading indicates the location of air parcels propagated using the ICON wind fields from the simulation initialised at 12 UTC 06 May 2013 and the orange contours those using the ICON wind fields from the simulation initialised at 00 UTC 07 May 2013. The blue triangles indicate the location of near-by radiosonde release locations. The dark purple shading shows WCB outflow up to 24 h after end of ascent and the light purple WCB outflow of 24 h to 48 h age (diagnostics based on ERA5). (b) Specific humidity as function of potential temperature at 00 UTC on 07 May 2013 from three radio sounding sides (red) and the ICON model simulations sampled over the origin region of the trajectories (shown in the left panel). Black and green colors correspond to ICON initialisation dates of 12 UTC 06 May 2013 and 00 UTC 07 May 2013, respectively. Note both simulations have been sampled in the trajectory origin region form the simulation initialised at 00 UTC 07 May 2013. The figure with profiles sampled over the respective origin regions from each simulation is shown in Fig. A3. The solid line shows the median profile, the shading indicates the $5^{\text{th}}$ to $95^{\text{th}}$ percentile range.

in humidity above 330 K. It is unlikely that this structure influenced the observed ExTL cirrus due to it being at a much higher isentrope. It may arrise due to the north-east advection of the radiosonde during its ascent and a weak moist filament ahead of the main gradient over Central Switzerland visible in the MSG data. However, in general the vertical profile information confirms the modelled UTLS humidity and temperature structure in the air mass origin region, in particular a large east-west gradient in UTLS humidity and relatively moist conditions around the tropopause in the air mass origin region. Finally, we compare the two ICON simulations initialised on 12 UTC 06 May 2013 and 00 UTC 07 May 2013. As discussed in sec. 3.3, the former fails to reproduce the observed ExTL cirrus and has a much higher tropopause in the measurement region than observed, analysed in ERA5, and modelled by the other ICON simulation. Trajectories arriving in the North Sea measurement area have a broadly similar path in the preceding 14-16 h both in terms of horizontal and vertical propagation: Both sets of trajectories originate over Southern Germany and experience a slow lifting while travelling north (compare Fig. 7 and Fig. A5). However, the cirrus cloud forms much later, with lower ice water content, and much smaller horizontal and vertical extent in the simulation



initiated on 12 UTC 06 May 2013 . This is consistent with a slight westward shift in the air mass origin into the region of dry
upper tropospheric air (as seen e.g. in the satellite data, Fig. 10 a). However, even if one assumes the same origin region in
both data-sets the ICON simulation with the earlier start date has lower specific humidity values between 318 K and 320 K
compared to the second ICON simulation and also compared to the Stuttgart radiosounding (Fig. 11 b). As the ExTL cirrus
forms in this altitude range, the delayed formation of more tenuous cirrus cloud is not surprising. Interestingly the evolution of
the ExTL and diabatic PV tendencies differs strongly between the two simulations (compare Fig. 9 and Fig. A6): The ExTL
(2-6 pvu) is about 1 km deep about 15 h before the air mass arrives in the measurement area. However, while it becomes
substantially less deep in the ICON simulation initialised on 00 UTC 07 May 2013, the geometric ExTL depth decreases
only slightly in the ICON simulation initialised on 12 UTC 06 May 2013. The decomposition of the diabiatic PV tendencies
into contributions from different physical processes suggests that the main difference is a less vertically focused radiative
PV production (at $z = \approx 10.0$ to $10.5\,\mathrm{km}$ and $t \approx -10\,\mathrm{h}$ to $-4\,\mathrm{h}$) and simultaneously stronger turbulent PV destruction (at
$z = \approx 10.5$ to $11.0\,\mathrm{km}$ and $t \approx -10\,\mathrm{h}$ to $-4\,\mathrm{h}$). These changes are consistent with the absence of an ExTL cirrus. Note, that
the optically thick cirrus layer in the upper troposphere is present in both simulations, although the trajectories destined for the
North Sea measurement region only move above this cirrus deck at later times (at $t \approx -10\,\mathrm{h}$ compared to $t \approx -12\,\mathrm{h}$) and over
shorter time period (until about $t \approx -7\,\mathrm{h}$ compared to $t \approx -4\,\mathrm{h}$) (Figs. 9 b and A6 a). In Fig. 12 we compare the trajectory path,
for those air masses arriving between 10.5 km and 11.0 km in the North Sea area, as well as the PV and wind field at 10.5 km
between the two simulations. Interestingly, from about 7 UTC ($\approx 7 - 9\,h$ before arrival in the measurement area) a negative
PV anomaly develops, which goes along with a stronger easterly wind component (anticyclonic rotation) in the simulation
initialised on 12 UTC 06 May 2013. While the two trajectory bundles propagate almost parallel to each other in the early part
of the simulation they converge into the North Sea measurement area in the final hours consistent with the wind field difference
(explaining the more westerly source area). Given the overall better agreement of the ICON simulation initialised on 00 UTC
07 May 2013 with the Learjet/TOSS observations, MSG satellite data, and radiosonde profiles over Southern Germany, one
may speculate whether the difference between the simulations provides evidence for the (local) importance of radiative PV
modification for the flow within the ExTL and the (thermo-)dynamical vertical structure of the ExTL.





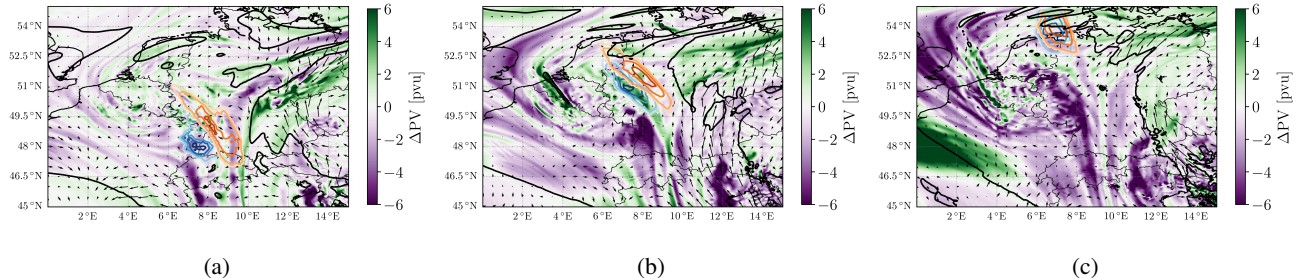

**Figure 12.** Difference in PV at 10.5 km altitude between the ICON simulations initialised at 12 UTC 06 May 2013 and 00 UTC 07 May 2013. The three panels correspond to valid times (a) 1 UTC, (b) 7 UTC and (c) 13 UTC on 07 May 2013, respectively. The arrows indicate the difference wind field at the same altitude. The contours show the position of back trajectories from the measurement area calculated with the wind field data from the 00 UTC 07 May 2013 (red) and 12 UTC 06 May 2013 (blue), respectively.

## 6 Summary and conclusions

The extra-tropical UTLS is a transition region (ExTL) between troposphere and stratosphere, which are characterised by very
different chemical composition as well as very different thermodynamic and dynamic properties. The ExTL structure, its spatio-temporal variability and long-term changes therein are important for our understanding of exchange processes between the troposphere and stratosphere (STE) as well as the climate state in general. Due to large potential temperature and potential vorticity gradients exchange between stratosphere and troposphere at the extra-tropical tropopause has to be facilitated by diabatic processes. Cirrus clouds frequently occur in the tropopause region and have even been reported to exist in the lowermost
stratosphere. Due to the associated latent heating, modified radiative transfer and turbulence characteristics, cirrus clouds may be one key component shaping the ExTL structure and STE. In this study, we combine observational data from various platforms with high-resolution model simulations and Lagrangian diagnostics to gain insight into the formation of a cirrus cloud observed over the North Sea in chemically and dynamically stratospheric air (Müller et al., 2015) and its impact on the ExTL structure. This specific case of an ExTL cirrus was observed by aircraft based in situ measurements during the AIRTOSS-ICE
campaign in May 2013 and was described earlier in Müller et al. (2015).

The AIRTOSS-ICE measurements allow for a unique characterisation of the thermodynamic structure of an ExTL cirrus due to the cloud being probed in situ with a dual platform approach. The observations on a second platform below the aircraft allow us to derive the vertical gradient of potential temperature from measurements as well as its change at the tropopause and in the region of ExTL-cirrus. Based on these observations we found a significant difference of gradients in potential temperature
inside and outside the cirrus (sec 3.2). Inside the cirrus we found weaker (neutral) stability compared to the surrounding, which indicates an influence of the cirrus on the thermodynamical structure of the sampled air masses in the ExTL. This finding could be reproduced in the model simulations, although the simulated gradient of potential temperature inside the cirrus was slightly higher than in the measurements (sec 3.3).

High-resolution model simulations ($\Delta x \approx 3.2\,\mathrm{km}$, $\Delta z \approx 150 - 200\,\mathrm{m}$ in the ExTL) suggest that the disturbed temperature





profile can be explained by radiative cooling towards the cirrus top and weak latent heating within the cirrus. Further, the La-
grangian PV diagnostics show a strong contribution of radiative cooling to the ExTL PV structure in the upper part of the cirrus
(sec 4.3). This contribution persists throughout the lifetime of the cirrus cloud and overall leads to a significant increase of
potential vorticity for trajectories passing close to the cirrus top. Further inside the cirrus layer, however, microphysical (latent)
heating causes a (small) decrease in PV consistent with a weakening stratification. Combined with smaller, but still substantial,

PV modifications due to turbulent momentum and heat transfer, the radiative PV modification causes a sharpening of the PV
gradient in the considered region of the ExTL (sec 4.3), diminishing the geometric distance between the 2 pvu and 6 pvu iso-
surfaces. However, only a small fraction of trajectories transitioned from PV values smaller to such larger than 2 $pvu$ during
the considered time period ($\approx$ 15 h prior to in situ measurments, sec 4.3). There is no indication in the model, that the ExTL air
mass gained additional moisture during this transit, making substantial STE during the transit unlikely. The model rather indi-

cates already relatively humid ExTL conditions in the origin region over Southern Germany. High observed relative humidity
in close vicinity to the cirrus in chemically stratospheric air is consistent with the already moist ExTL air mass. Moistening
through sublimation of the cirrus particles is therefore not required or even unlikely (sec 3.2, 4.2). Satellite measurements and
radiosonde data confirm the presence of an already moist air mass in the formation region extending into the ExTL. Consistent
with the satellite data, the model suggest a persistent existence of cirrus particles since their formation over Southern Germany

around 12 h before the observation, albeit with varying ice content. During the transit to the measurement area, the air mass
was subject to gravity wave activity over central Germany and was steadily lifted by up to 1000 m (sec 4.2, 5). The formation
conditions match the slow updraft type of in situ cirrus.

Note that these results slightly differ from the results by Müller et al. (2015), where trajectory analysis indicated ice particle
formation slightly before the measurements during slow ascent to the measurement region. However, these results were de-

duced from a coarser analysis data set. Satellite data are consistent with an earlier cirrus formation as seen in our model data,
but there remains some uncertainty regarding the representation of ice nucleation via deposition nucleation and homogeneous
nucleation in the ICON model. The model may have to active deposition nucleation, which is expected to favor earlier cirrus
formation to (slightly) earlier times for the thermodynamic evolution seen along the trajectories in our study. Note further,
that the in situ gradient information as measured by the TOSS are new in our study. The decrease in stratification inside the

cirrus layer is less pronounced in the model than in the observations. Shallow cirrus convection as proposed by Spichtinger
(2014) could be a plausible process responsible for the stronger redistribution of potential temperature since the vertical model
resolution does not allow for the explicit representation of convection at these small scales. Neither does the vertical model
resolution allow the representation of the strong vertical temperature gradient at the cloud top as evident in the observation due
to only 2-3 model levels in the relevant area.

Regarding the diabatic impact of the cirrus layer on the tropopause structure we found a strong contribution from radiation
on the PV change for our case even higher than the impact from turbulence. These results differ from Spreitzer et al. (2019)
who found a stronger contribution by turbulence. In our case trajectories indicate very high values of humidity in the lower
stratosphere 12 h prior to the measurements with high values of relative humidity with respect to ice also outside the cirrus
layer. It is noteworthy, that the cirrus particles persisted during the transit in (over)saturated air while only a few of the corre-





sponding trajectories changed from tropospheric to stratospheric PV values. The model does not indicate cirrus particles in the ExTL at the start of the simulation, which implies the in situ formation of the ice particles in the ExTL during the transit time. The conditions permitting cirrus formation (i.e. sufficient humidity) were present in this particular region of the ExTL. The source process that transported the high amounts of humidity into the lower stratospheric layer where the cirrus particles were measured could not conclusively be identified. A model simulation with an earlier start date failed to reproduce the ExTL cirrus

cloud and therefore did not allow to trace the air mass history over a larger time horizon. In the shorter simulation, the origin of the comparatively high amounts of water vapor could be traced back to an already humid air mass around the transition layer prior to the measurements. A diagnostic based on ERA5 suggests an influence of WCB outflow in the source region hours before the observation, which provides a plausible explanation for the increased humidity and the long stratospheric residence time of cirrus particles before the measurements. Satellite data point to convective activity over Eastern Germany in the 36 h

preceding the observations, which could further impact the ExTL humidity in the source area.

    Our analysis might hint towards a relevant process for cross-tropopause exchange and diabatic PV-modification, which has been underestimated so far due to the limited capability of models and analysis data to resolve the underlying processes.





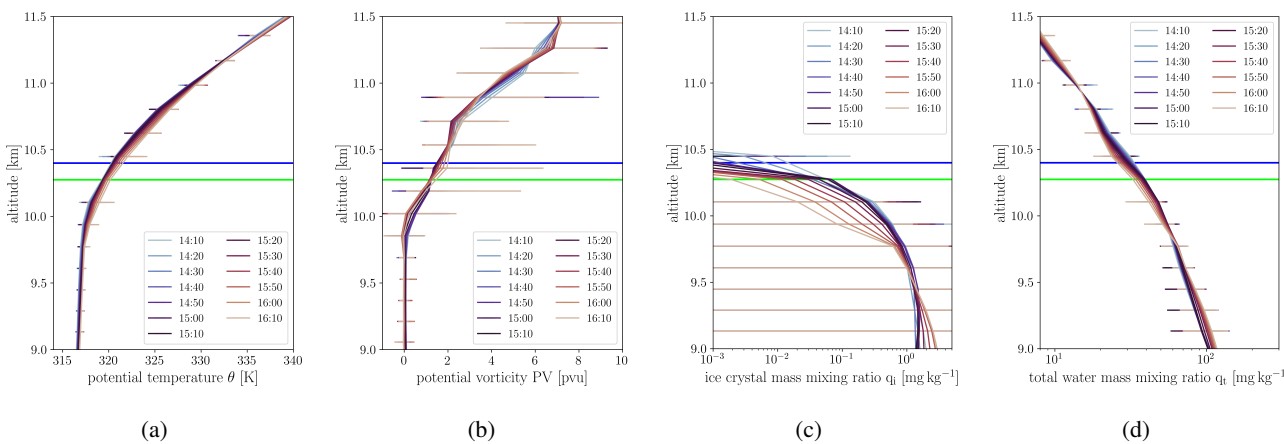

**Figure A1.** Height profiles of the average (a) potential temperature, (b) potential vorticity, (c) total water mass mixing ratio, and (d) ice mass mixing ratio from the ICON simulation. The different line colors represent profiles sampled at different times between 1410 UTC and 1610 UTC on 07 May 2013. Averages are taken over all grid points within $6.5 - 7.5\,°E$ and $54.3 - 55.1\,°E$; spatial variability as represented by the maximum and minimum value at a particular altitude are shown in the bars. The average altitude of the Learjet leg discussed in section 3.2 is indicated by the dark blue horizontal line and that of the TOSS by the green horizontal line. The simulation starting at 12 UTC on 06 May 2013 is shown.







**Figure A2.** MeteoSat Second Generation (MSG) satellite images at 10.8 $\mu$m. The observation time increases from top left to bottom right from 00 UTC to 13 UTC on 07 May 2013. The contours indicate the position of back trajectories from the measurement area based on the simulation initialised at 00 UTC 07 May 2013 (orange) and 12 UTC 06 May 2013 (green), respectively.



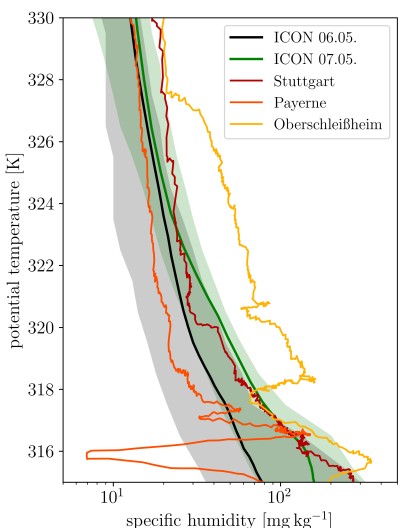

**Figure A3.** Specific humidity profile as function of potential temperature at 00 UTC on 07 May 2013 from three radiosonding sides (red) and the ICON model simulations sampled over the origin region of the trajectories (shown in the left panel). Black and green colors correspond to ICON initialisation dates of 12 UTC 06 May 2013 and 00 UTC 07 May 2013, respectively. In contrast to Fig. 11 the simulations have been sampled in their respective trajectory source areas. The solid line shows the median profile, the shading indicates the $5^{th}$ to $95^{th}$ percentile range.



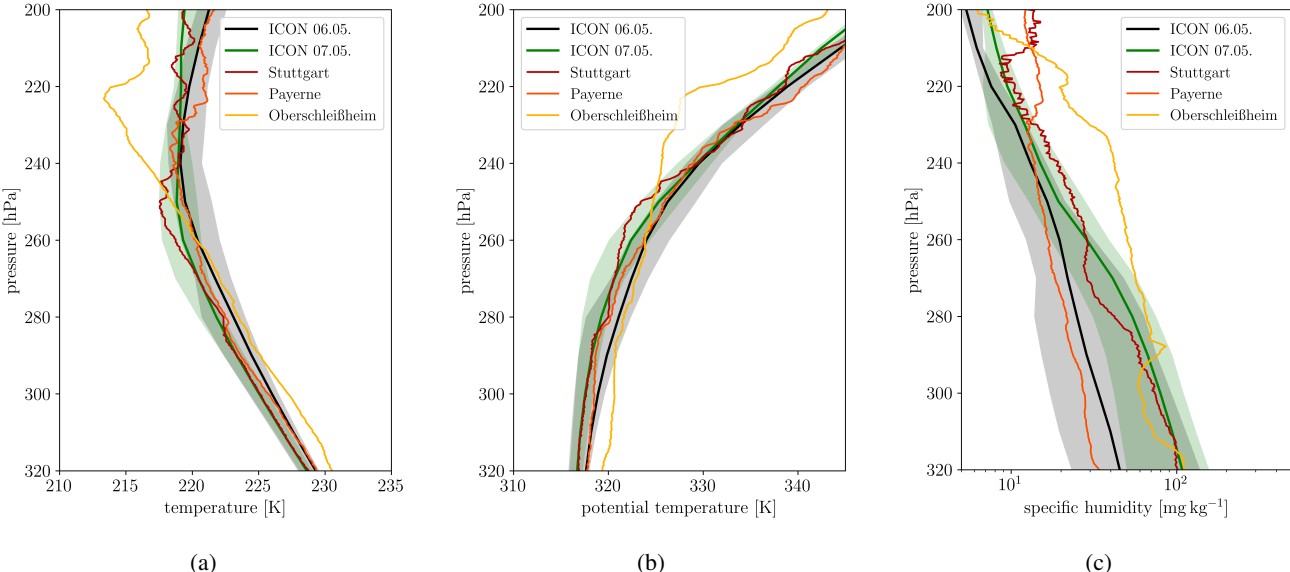

        (a)                             (b)                             (c)

**Figure A4.** (a) Temperature, (b) potential temperature, and (c) specific humidity as function of pressure from three radio sounding sides (red) and the ICON model simulations sampled over the origin region of the trajectories (shown in Fig. 11). Black and green colors correspond to ICON initialisation dates of 12 UTC 06 May 2013 and 00 UTC 07 May 2013, respectively. In contrast to Fig. 11 the simulations have been sampled in their respective trajectory source areas. The solid line shows the median profile, the shading indicates the $5^{\text{th}}$ to $95^{\text{th}}$ percentile range.



**Figure A5.** Path of trajectories arriving in measurement area between 13:40 UTC and 16:00 UTC with: (a,b,c) tropospheric marker, (d,e,f) stratospheric marker, and (g,h,i) pv $\geq 2$ pvu and $q_i \geq 10^{-3}$ mg kg$^{-1}$. Simulation starts at 12 UTC 06 May 2013.





(a)

(b)

(c)

(d)

(e)

**Figure A6.** Composite profile along air parcels arriving in the measurement area with $pv \geq 2\,pvu$ and $q_i \geq 10^{-3}\,mg\,kg^{-1}$ from the simulation started at 12 UTC 06 May 2013: Variables shown are (a) PV, (b) PV tendency from radiation parameterisation, (c) PV tendency from latent heating by turbulence parameterisation, (c) PV tendency from microphysics parameterisation, and (d) PV tendency from momentum transport by turbulence parameterisation. The columns have been retrieved from 10 min Eulerian output at the native ICON grid and have been interpolated to the horizontal location of the air parcels.



*Code and data availability.* The observational data from Learjet and TOSS as well as derived time series will be published on zenodo. The main model data used in this paper as well as all programs used for postprocessing and visualising model data will also be published
on zenodo upon acceptance of the paper. Before this please contact the authors to obtain access to the code. The ICON model code is not published, as the model version used here predates the open source release of ICON. The radiosonde data from the German stations Oberschleißheim, Stuttgart, and Norderney was obtained from opendata.dwd.de. The high-resolution radiosonding data from Payerne was kindly provided by Philipp Bättig from MeteoSwiss. MSG satellite data have been accessed through EUMETSAT. IFS analysis data for model initialisation was obtained from ECMWF.

*Author contributions.* NE analysed the observational data, AM conducted and analysed the ICON simulations. All authors contribute to the
design and writing of the manuscript.

*Competing interests.* At least one of the authors is a member of the editorial board of ACP.

*Acknowledgements.* Funded by the Deutsche Forschungsgemeinschaft (DFG, German Research Foundation) – TRR 301 – Project-ID 428312742, projects B01, B08, Z01. The authors gratefully acknowledge the computing time granted on the supercomputer MOGON 2
at Johannes Gutenberg University Mainz (hpc.uni-mainz.de), which is a member of the AHRP (Alliance for High Performance Computing in Rhineland Palatinate, www.ahrp.info) and the Gauss Alliance e.V. Fig. 10 and Fig. A2 contain modified EUMETSAT Meteosat Second Generation HRSEVIRI data. The native HRSEVIRI data has been download via EUMDAC in February 2024. Philipp Bättig from MeteoSwiss kindly provided the high-resolution sounding data from Payerne. Hans-Christoph Lachnitt provided the interpolated ERA 5 PV data at the Learjet position.



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
