# Peer review of "Impact of cirrus on extratropical tropopause structure"

_EGUsphere, 2024_

## Referee Comment (RC2)

Review of "Impact of cirrus on extratropical tropopause structure" by Nicolas Emig, Annette Miltenberger, Peter Hoor and Andreas Petzold

The paper describes in detail the observation and formation of a cirrus cloud in the extra-tropical transition layer (ExTL) over the North Sea by means of observations that have been taken during the AIRTOSS-ICE campaign and high-resolution ICON simulations including calculation of online trajectories. The combination of the observational data and the high-resolution model simulations enables to investigate in detail the formation of the cirrus cloud and its impact on the ExTL structure. Thereby, the model performance in simulating this case is discussed and the impact of diabatic processes on the modification of the temperature and potential vorticity profile in the ExTL is determined based on temperature-tendency output from the model simulation. The methods that are presented are appropriate and the study provides new insights in how cirrus clouds can influence the ExTL. The paper is therefore suitable for publication after minor revisions.

General comments:

The paper is generally well written, and the main findings are supported by the figures. However, some of the figures and the associated text need some clarification. Please also read carefully all your figure captions because in several figures they are not complete, and some necessary description is missing.
For more details see my comments below.

Abstract: I would remove the sentence "Earlier analysis by Müller et al. (2015)" here and instead emphasize that you present a combination of measurements and a high-resolution simulation including a Lagrangian analysis. Therefore, I would slightly reformulate the first paragraph of the abstract to something like:

Here we present for the first time a combination of in situ observations of the ExTL thermodynamic structure in- and outside cirrus by utilizing a dual-platform approach and a Lagrangian analysis based on high-resolution simulations. The observational data were collected during the AIRTOSS-ICE campaign.
The dual-platform approach reveals substantial disturbances in the vertical profile of potential temperature with a weakened stratification inside the cirrus and sharpening above. ...

L 34: ...isentropic composition gradients indicate the effect of irreversible transport .... Why is this the case? Can you briefly explain?

L 121 and 124: ICH-Sensors → MCH-sensors ...?

Fig. 1: "colour bar"; to what exactly do you refer to here with LOW and HIGH?

L164: ...form in the the humid -> delete "the"

L172/173 → .... **dark** grey for the TOSS-platform and light grey ....

L175 and Figure 2: could you please indicate in Figure 2 on which section you focus?

Figure 2: the figure does not provide important information; I find the information given in Figure 3 much easier to understand and more important. I would either remove Figure 2 or maybe show additionally a map below the flight pattern such that it becomes clear where the Learjet is flying, also to be able to compare it to the satellite images.

Figure 3 a, theta gradient: wouldn't it be more interesting/intuitive to see the difference in theta values between the Learjet and Toss measurements, because you also state that the vertical distance deltaz between Toss and Learjet was constant (thus, K/deltaz), instead of showing a gradient in units of K/km? A gradient of 25 K is huge and doesn't tell us anything about the stratification in the considered layer? Or do I misunderstand something here? Thanks for clarifying.

L 200: ...6.6 °° E → 6.6 °E

L 200: One of the two mixing lines showing .... → which one? What does the other show?

L 200 – 207: I have difficulties to follow your description of Figure 4. Can you describe in more detail what exactly the two different mixing lines you are referring to are? You also say that RHi reaches 100% inside the cirrus layer? However, the values inside cirrus are already blue, so supersaturated? Thanks for clarifying.

Fig. 5 d: x-axis, insert missing bracket

L 241: Over the analysed 2 h time window.... Shift this sentence to line 235, thus to description of figure 5.

Figure 6: missing bracket in x-axis description of panel b); figure caption last words: ...and no clouds (red) being present. In general, you could think about different colours for the red and green lines due to colour blindness.

L 242-249: you may want to shift the description of figure 5d to line 236 before you start the discussion of figure 6.

L 258: to investigate these processes we analyse...

Figure 7: what are the vertical dashed lines? Why do the trajectories extend beyond the target region? Improve colour scale for panels a, d and g, because most of the values are in the green colour range.

L 293: histograms in Fig. 8a

L 300-301: considering the parcel's initial PV .... What exactly do you mean here? Why is it important here and what does it tell us?

Section 4.3: This section, including Figure 9 contains interesting and important information. However, the explanation in the text and the link to the figures are sometimes not easy to follow. So please make sure that all lines, hatching, dots, .... are explained in the figure caption of Figure 9 and refer to the plots in the text (including, white solid/dashed line, grey circles, dots, red hatched area, green solid line). For example, it would be very helpful to have the 2 pvu and 6 pvu isolines in each panel (are the white lines in panel b the pv isolines?). In panel b I would also improve the colour of the PV values. Now it is not easy to see the values precisely, one can only estimate the value at a given point.

L 360: 8,7 → 8.7

L 365: ...more tenuous cirrus further east (Fig. 10c). I have difficulties to see where the tenuous cirrus are, how they appear in panels c, f, i. Some more explanation would help.

L 389: You state that the moist layer is not as sharply capped in the model as in the observations. However, I cannot see this in the lines in Fig. 11b.

Figure 11: I would use the same colours for the triangles indicating the radiosonde stations and the lines in panel b). For example, the triangle for Oberschleissheim should be yellow. I would also either remove the WCB outflow, because this is not mentioned in the text describing Figure 11 or add the text mentioning the importance of the WCB outflow that is in the summary (L492) also here.

L420: ...a negative PV anomaly develops. Is it really a negative PV anomaly or is just the difference negative? In general, it is difficult to relate the text to the figure. Can you please improve this and point more precisely to the regions you are referring to.

L429: ...., which is characterised by

L459: I don't understand what you mean here with "making substantial STE during the transit unlikely". Where does this conclusion come from?

---

## Author Response (AR1)

**Impact of cirrus on extratropical tropopause structure - review comments**

**April 23, 2025**

We thank the reviewers for their careful reading and the stimulating comments, which helped to improve the paper. We tried to address carefully their criticism and give below our comments with the reviewers statement in black, our comments in orange and our actions or statements in blue.

**1 Referee 1**

**1.1 Summary**

This paper looks at cirrus clouds that formed in the extratropical transition layer using observations and model simulations. The authors look at the case study as Muller (2015), but include observations from a second set of instruments towed by the aircraft, in addition to the aircraft observations, which allows them to show the different gradients in and out of the cirrus clouds. The authors also run high-resolution simulations to determine the origin of air and the physical processes responsible for forming the cirrus and modifying the gradients in the extratropical transition layer. They show that radiation and turbulence are important for increasing PV above the cirrus, and microphysics for decreasing PV below.

This is an interesting study with interesting results but needs some clarification of figures and discussion before publication.

**1.2 Comments**

- All figures
  - There is an inconsistent font size and it is sometimes quite small and difficult to read. I assume this is because you have images of various widths that are being resized. Please make the figures to a consistent width, so that the font size is consistent and readable.
  - The labelling of panels below each panel is confusing at first. A particular example is Figure A2, where the labels look like they correspond to the panel below.

Thank you for your comment, we have carefully looked at all figures and rectified these issues. We have also shifted the figure label position to top-left to enhance readability.

• Section 2.1. For someone not intimately familiar with observational equipment, this was difficult to follow. There are a number of undefined acronyms, although I don't think it is important to spell out the acronyms as long as the names of the instruments are attached to a citation or an explanation of what they do. FSSP, ICH, and MCH are not explained. This might be better presented as a table with columns for: quantity measured, sensor name, sensor reference, platform (Learjet or TOSS), sensitivity.

MCH (MOZAIC Capacitive Hygrometer) and ICH (IAGOS Capacitive Hygrometer) are a type of sensor used in the MOZAIC project and later in the IAGOS project respectively. The difference between these two is documented in (Neis et al. 2015b). The sensors used during AIRTOSS ICE (this study) are ICH.

We changed each instance of "MCH" in the manuscript to "ICH".

FSSP stands for Forward Scattering Spectrometer Probe.

We added the name and particle size range as well as a reference to the manuscript.

We also added the particle size range for the CCP instrument.

• Similarly in section 2 there needs to be explanations or references for what the IFS is and what R3BX means. IFS (integrated forecasting system) is the name of the ECMWF numerical weather prediction model. R3BX

Figure 1: (a) Geographic location of the flight path (orange line) on 07 May 2013. Colours: pressure on 2 pvu isosurface. The measurement area  $(6.5^{\circ} - 7.6^{\circ}\text{E} \text{ and } 54.3^{\circ} - 55.1^{\circ}\text{N})$ , yellow box) is shown in Fig.1 b) in Detail. (b) Flight path for the analyzed flight limited to the measurement area. The flight section considered in this analysis is highlighted in purple. Cirrus occurrence measured at Learjet (TOSS) is marked in yellow (red).

describes the topology of icosahedral grid used in the ICON model. The key information for persons not familiar with the icosahedral grid of ICON is the effective grid spacing which is given in brackets behind the name of the grid.

We changed "IFS operational analysis" to "ECMWF (IFS) operational analysis" and altered the text to make it more clear that R3BX refers to the name of the grid.

• Can you mark the start and end point of the leg you are analysing on figure 2. It's not clear from the text exactly which part you are talking about. I think it is the roughly straight section from (6.7E, 55.05N) to (7.3E, 55N), given the discussion of figure 4, but I'm not sure. I then struggled to line this up with 3a. In figures 2 and 3a, there is a section with both Learjet and TOSS in cirrus, but in figure 2 this is surrounded by TOSS being continuously in cirrus, and in 3a TOSS is discontinuously in cirrus or there are no measurements?

We agree that Fig. 2 was lacking clarity. To clarify what is happening where we reworked it (See Fig. 1 in this document):

We added a panel with the whole flight path on a map for geographic context. This panel also includes a box that corresponds to the "measurement area" (6.5 - 7.5 E, 54.3 - 55.1 N) which is later used for the Lagrangian analysis. The second panel shows the flight path inside the measurement area and highlights the analyzed section and cirrus occurrence. The potential temperature colour coding from the previous version of this figure was redundant (Fig. 3 includes all information about potential temperature) and is now omitted to make the figure more clear.

Fig. 3: The cirrus is continuous, but the theta-measurements on the TOSS are not. The time series of Theta is marked with cirrus occurrence and has gaps.

- You state that the TOSS has a constant theta of 320K, but it shows about as much variation as Learjet. We rephrased the statement:
  - "...the potential temperature at the TOSS varies around  $\theta = 320$  K."
- Figure 4. There are a lot of overlapping points in this figure, so it's not clear whether the colour show is representative of the average or just the last point plotted. It might help to split each panel into two figures (one for each leg/distinct air mass), which would also back up the discussion which says that one mixing line corresponds to a different flight leg.

The colours of the respective points represent just one measurement. The representation as a tracer-tracer-correlation, while advantageous for the interpretation of dynamic regimes, can obscur information for data points with similar tracer mixing ratios.

To reduce this problem, we reworked the figure slightly with smaller datapoints and different markers for cirrus

Figure 2: As in manuscript, but with reduced point- and marker size to minimize overlap of points.

measurements (See Fig. 2 in this document).

- L314 It would be good to define the "measurement area" here. I assume it is the grid point range described in figure 5, but it would be better to also say it in the text. Also add a visual reference, such as the box in Figure 7, to figure 2 to give context to the size of the area in comparison to the flight leg.

  We improved the graphical representation of the "measurement area" by changes to Fig. 2 as described above: Flight track and measurement area are depicted there on a map with the same borders that are used in the model analysis. This figure is now referenced in L214 (old manuscript).

  In addition, we added the geographical extent of the "measurement area" used to extract data from the model.
- L227 "This is in slight contrast to ERA5 derived PV in Fig. 3". It's hard to say if the model and ERA5 are actually inconsistent based on comparing figures 3 and 5. Figure 3 is exactly co-located with the flight track, whereas figure 5 is for all points in a box and the horizontal lines show substantial variability in PV at that level. We agree that this conclusion is not straightforward from comparing Fig. 3 and 5. However, we have also interpolated ICON onto the aircraft position and this shows values between 2 and 3 pvu along the entire flight track in contrast to the ERA5, which shows values between 2 and 4 pvu. As we do not want to show the interpolated ICON data due to the already large number of figures in the manuscript, we have altered this sentence as follows:

This is consistent with ICON data interpolated to the Learjet position, which shows PV values in the range of 2-3 pvu (not shown), which is slightly lower than in ERA5.

- L235 "Notably, the dynamical tropopause in the model is about 500 m lower than the cloud top suggesting the presence of ExTL cirrus consistent with observations". This is based on box averages so isn't clear evidence. Particularly when you show in figure 6c that air parcels with cirrus on average have lower PV around the tropopause level (a higher tropopause). I would suggest to delete this sentence, because the following discussion of figure 6 showing regions with ice and PV>2 is more convincing
  - We agree that Fig. 6a. produces more convincing arguments for the existence of an ExTL cirrus in the model. We therefore removed the sentence.
- Figure 6a Similarly to figure 4, there are many points overlying each other, so it is not clear whether the colour is in any way representative and what sort of density of points occupy each region.

  As we do not discuss the information on stratification, which is indicated by the colourscale, in the text, we have changed to a 2D histogram figure, which conveys the same information in a more quantitative manner.

• Figure 6 b/c – There are two different blue lines in figure and legend not mentioned in the caption. Is the cyan line a subset of the blue line, for points with PV>2. The lines appear identical. Does clouds below 10km mean cloud tops below 10km or are there grid points included in both the blue and green lines where the clouds go across 10km?

The cyan line is indeed the sub-set of the profiles included in the calculation of the blue line. However, there are only about 5 % of the profiles with cirrus top above 10 km and maximum in-cloud PV smaller than 2 pvu. Therefore the blue and cyan line are almost identical. The cloudy profiles are sorted according to cirrus top, which means there is no overlap between the profile data used to compute the blue and green lines.

To remove the potential confusion, we removed the cyan line from the figures. In the caption we also state now that cirrus cloud top is used for categorizing the profiles.

• Fig. 7 – As with figures 4 and 6, the overlapping trajectories means the colour in the figure is probably not representative. The idea to focus on model trajectories that are stratospheric and have cirrus, rather than exactly co-locating model data with observed cirrus seems entirely justified anyway, so I would suggest removing Figure 7 and the discussion in section 4.1

Fig. 7 shows the general behaviour of the different subsets of trajectories in the simulation (it also gives a nice overview of the time, place and lifting). It also motivates the selection of data for further analysis:

- 1) The results of the simulations are consistent with the different airmasses as divided by the LearJet-measurements.
- 2) Given 1) we can generalize the results and are no longer restricted to airmasses sampled by the LearJet. Now we can use the PV > 2pvu,  $qi > 10^{-3}mg/kg$  criteria. This gives us more trajectories to work with which still represent the relevant processes.

However, we agree that the overplotting may be misleading in some places (or at least mask whether the data is representative).

To mitigate theses issues we now only show every tenth trajectory and have reduced the linewidth.

While this not fully prevents overplotting, we think the new figures nonetheless provide an essential insight into the general horizontal and vertical pathway of the trajectories. This information is vital for the analysis in the manuscript. We think this information is conveyed despite in some places there are too many trajectories to see all the data-points.

• L285 - "The extensive cirrus cloud seen in Fig. 7 emerges in the 12 h before the arrival of the trajectories in the measurement area and arises by a combination of moist ExTL air over southern Germany and strong lifting of up to 1000 m (median: ≈ 500 m) during its northward propagation." − What is this based on?

Fig. 7 shows the loss of pressure, but indeed not the lifting in terms of altitude. However, we provide numbers for the latter as these are more common in the cloud community. The numbers have been computed based on the trajectory data. Also this information can be seen in Fig. 9, which we do not want to cite before it is actually introduced in the text. Also the information that the air is moist already over southern Germany is only indirectly shown in Fig. 7 (more vigorous evidence follows in Fig. 8 and Fig. 11). However, the early appearance of cirrus suggests that the air is already quite moist from the start. We have rephrased the sentence to make it clear that this statements is speculative given Fig. 7, but draws on information presented later in the manuscript.

The extensive cirrus cloud seen in Fig. 7 emerges in the 12 h before the arrival of the trajectories in the measurement area and arises by a combination of moist ExTL air over southern Germany (see discussion of Fig. 8 and in section 5) and strong lifting of up to 25 hPa, which corresponds to a vertical displacement of about 1000 m (median:  $\approx 500$  m) (not shown), during its northward propagation.

• Figure 8 – What does "substantial accumulated ice nucleation rates" mean specifically? (Same for L292). Depending on what you mean by this, this could account for your "not shown" conclusion on L289

The threshold that has been used for Fig. 8 is  $10^{-8}$  kg kg-1 accumulated ice formation from nucleation processes over the entire length of the trajectories, which is about 15 h on average. Changing this to  $10^{-9}$  kg kg-1 quantitatively changes the results by including more points in the range 80-100;%  $RH_i$  (Fig. 3 a). Lowering the threshold (tested up to  $10^{-13}$  kg kg-1) further does not change the results further. Fig. 3 b shows the distribution of accumulated nucleation rates across all trajectories. However, given that in the ICON model ice nucleation is not possible in ice supersaturated conditions, we argue that the threshold of  $10^{-8}$  kg kg-1 is adequate and any trajectories with marginally lower accumulated nucleation rate are likely passing in close vicinity of the cloud edge and "inheriting" small nucleation rates due to the interpolation of grid-scale variables to the trajectory position.

In L298 no threshold has been applied. The nucleation rates from all ice formation process except deposition nucleation are exactly zero along the trajectories. We therefore also do not show this as this would be an increasingly boring figure.

We have added the information on the threshold in the caption of Fig. 8. And modified the sentence in L289 as

Figure 3: (a) Distribution of  $RH_{ice}$  values in cirrus clouds forming along the trajectories arriving in the measurement area. The dark blue histograms shows the distribution, if only parcels substantial accumulated ice nucleation rates larger than  $10^{-9}$  kg kg are taken into account. Fig. 8a in the manuscript contains the same figure, but showing in dark blue the distribution of  $RH_{ice}$  for trajectories with accumulated ice nucleation rates larger than  $10^{-8}$  kg kg. (b) Distribution of accumulated ice nucleation rates along all trajectories containing an ExTL cirrus in the measurement region.

follows: "Cirrus clouds are formed in the model by deposition nucleation as indicated by the process rates from the different ice formation parameterisations in ICON traced along the trajectories , which are zero for all processes except deposition nucleation (not shown)."

- L300 "If considering the parcel's initial PV for constructing the vertical profile, the same structure emerges with small variations (not shown)". Why not show this as an extra panel in Figure 8?

  The figure using the initial PV for constructing the vertical profile is shown in Fig. 4 (this document). It conveys essential the same information, but is more difficult to connect to the observed profiles and the other profile information shown e.g. in Fig. 5, 6 and 9, which all focus on the PV distribution in the target area. As the manuscript already contains a lot of figures and there is no new information in this figure, we maintain our dicission to not show this figure in the manuscript.
- Figure 9 i) Explain the green line and red hatching showing in (c)-(f), and the white lines, red hatching, and dots in (b) in the caption. What is the difference between filled and unfilled hatches in (b)? ii) Why is the red hatching slightly different in (b) compared to (c)-(f)? iii) Figure 9(a) is used as evidence against troposphere-stratosphere exchange, but the grey shading only shows the 25-75th percentiles of PV, so there could still be a large amount of air parcels that fit the criteria. Also, as I understand it 9a shows the full set of trajectories that arrive in the measurement area, not just the cirrus subset. What fraction of these air parcels actually belongs to the cirrus subset?
  - i) We apologize, this information was missing from the caption.
  - ii) This was a mistake from us where the panels c-f did not show the most recent version of the trajectory data-set.
  - iii) All panels in Fig. 9 contain only trajectories that have a cirrus cloud in the measurement region and PV values larger than 2 pvu. The few points marginally below the 2 pvu in the profile are due to the vertical axis being constructed from the mean over the time segment, during which trajectories are in the measurement area while the selection is based on instantaneous values. If the full range is shown (Fig. 5, this document), there are indeed some trajectories that undergo STE: These are reflected in particular to all grey shaded areas that are left of the orange line. This accounts for 12.5% of the trajectories, but are constrained to altitudes close to the 2 pvu line (their mean and maximum PV in the measurement region are 2.24 and 2.89 pvu, respectively). Still, the statement holds that most of the cirrus trajectories do not undergo STE. If we consider the sub-set of trajectories that undergo STE after the formation of cirrus, i.e. PV<2 pvu at the first time at which  $q_i > 10^{-12}$ , the fraction reduces further to 7.67% of the trajectories (as was already stated in the original manuscript in L308). That means while some cirrus may be formed in the troposphere and later on make the transition to the stratosphere, this is not the case for the vast majority of data.

Figure 4: Change in parcel total water content  $\Delta q_t$  from the start of the simulation, i.e. 00 UTC 07 May 2013, until the parcel arrives in the measurement area. The black line shows the median  $\Delta q_t$ ; the grey shading the  $25^{\rm th}-75^{\rm th}$  and  $5^{\rm th}-95^{\rm th}$  percentile range, respectively. Similarly the change of total water from the turbulence parameterisation is shown in blueish colors and that from sedimentation into the parcels by warm colors. In contrast to Fig. 8b in the manuscript trajectories the vertical axis is constructed using the PV values of parcels on 00 UTC 07 May 2013 and not their PV values upon arrival in the measurement region.

Information on i) has been added to the caption of Fig. 9.

- ii) Panel c-f have been updated in Fig.9.
- iii) Some additional text and explanation along the lines outlined above has been added to the revised manuscript in L.310 ff.

**2 Referee 2**

**2.1 Summary**

Review of "Impact of cirrus on extratropical tropopause structure" by Nicolas Emig, Annette Miltenberger, Peter Hoor and Andreas Petzold

The paper describes in detail the observation and formation of a cirrus cloud in the extra-tropical transition layer (ExTL) over the North Sea by means of observations that have been taken during the AIRTOSS-ICE campaign and high-resolution ICON simulations including calculation of online trajectories. The combination of the observational data and the high-resolution model simulations enables to investigate in detail the formation of the cirrus cloud and its impact on the ExTL structure. Thereby, the model performance in simulating this case is discussed and the impact of diabatic processes on the modification of the temperature and potential vorticity profile in the ExTL is determined based on temperature-tendency output from the model simulation. The methods that are presented are appropriate and the study provides new insights in how cirrus clouds can influence the ExTL. The paper is therefore suitable for publication after minor revisions.

**2.2 Comments**

- General comments:
  - The paper is generally well written, and the main findings are supported by the figures. However, some of the figures and the associated text need some clarification. Please also read carefully all your figure captions because in several figures they are not complete, and some necessary description is missing. For more details see my comments below.
- Abstract: I would remove the sentence "Earlier analysis by Müller et al. (2015)" here and instead emphasize that you present a combination of measurements and a high-resolution simulation including a Lagrangian analysis. Therefore, I would slightly reformulate the first paragraph of the abstract to something like:

Figure 5: Change in parcel potential vorticity  $\Delta PV$  from the start of the simulation, i.e. 00 UTC 07 May 2013, until its arrival in the measurement area. The black line shows the median  $\Delta PV$ ; the grey shading indicates minimum-maximum range (In contrast Fig. 9a in the manuscript shows the  $25^{\rm th}-75^{\rm th}$  percentile range in grey.).  $\Delta PV$  due to temperature changes from the radiation, microphysics, turbulence, and orographic drag parameterisation are shown by the red, blue, dark green, and magenta lines (plus shading). The light green line (plus shading) shows  $\Delta PV$  from vertical momentum flux (turbulence and orographic drag parameterisations). The orange line indicates the combinations of  $\Delta PV$  and PV that signifies a PV value of 2 pvu at 00 UTC. Therefore all points to the left of the orange lines are parcels which cross the dynamical tropopause in the considered time period.

Here we present for the first time a combination of in situ observations of the ExTL thermodynamic structure in- and outside cirrus by utilizing a dual-platform approach and a Lagrangian analysis based on high-resolution simulations. The observational data were collected during the AIRTOSS-ICE campaign. The dual-platform approach reveals substantial disturbances in the vertical profile of potential temperature with a weakened stratification inside the cirrus and sharpening above. . . .

We thank the reviewer for the suggestion and followed it largely, but want to keep the link to the first study which stimulated our research.

We changed the abstract as follows: Here we present for the first time a combination of in situ observations of the ExTL thermodynamic structure in- and outside cirrus using a dual-platform approach during the AIRTOSS-ICE campaign and a Lagrangian analysis based on high-resolution simulations. Earlier analysis by Müller et al. (2015) suggests cirrus formation in stratospherically influenced air based on measured  $N_2O$  mixing ratios. The dual-platform approach reveals substantial disturbances in the vertical profile of potential temperature with a weakened stratification inside the cirrus and sharpening above.

• L 34: ... isentropic composition gradients indicate the effect of irreversible transport .... Why is this the case? Can you briefly explain?

We thank the reviewer for this point, since the statement was a bit too simplified. It holds for species as long as they can be regarded as passive relative to the time scale of isentropic mixing and of the underlying driving diabatic process. For the case without irreversible transport, one would expect a quasi-isentropic distribution of these trace gases. Therefore gradients of species on isentropic surfaces may be a potential indication for diabatic processes prior to the observation. Since this depends on the relation of chemical and dynamical time scales, such gradients potentially indicate this relation. In the manuscript, 1.35 ff. give some further explanation about diabatics.

We shifted the sentence to the start of the following paragraph and added the word 'potentially'.

L 121 and 124: ICH-Sensors → MCH-sensors ...?
 See also the answer to the other referee.
 MCH and ICH were just internal labels, the correct name is "ICH" (Neis et al., 2015a) - changed

• Fig. 1: "colour bar": to what exactly do you refer to here with LOW and HIGH?

The figure shows the principle of measurement qualitatively only. "HIGH" and "LOW" refer to values of an arbitrary quantity (e.g. potential temperature) measured by both platforms (colours correspond to the

background of the figure).

We added to the captions: Background colours represent an arbitrary air mass property changing from low to high values at the tropopause, which can be measured simultaneously by the two platforms.

- L164: ... form in the the humid → delete "the"
   Thanks! Deleted.
- L172/173  $\rightarrow$  .... dark grey for the TOSS-platform and light grey .... We reworked the figure this refers to (see also below) Changed to the appropriate colours in the reworked figure.
- L175 and Figure 2: could you please indicate in Figure 2 on which section you focus? We changed the figure (see also below) and added a sentence to the text "(see Fig. 2, the considered section is marked in purple)."
- Figure 2: the figure does not provide important information; I find the information given in Figure 3 much easier to understand and more important. I would either remove Figure 2 or maybe show additionally a map below the flight pattern such that it becomes clear where the Learjet is flying, also to be able to compare it to the satellite images.

Thanks for the suggestion. We modified Fig. 2 by omitting the redundant information about potential temperature and added a second panel with a map to put the location of the flight path in geographic context. (See Fig. 1 in this document).

- Figure 3 a, theta gradient: wouldn't it be more interesting/intuitive to see the difference in theta values between the Learjet and Toss measurements, because you also state that the vertical distance delta(z) between Toss and Learjet was constant (thus, K/delta(z)), instead of showing a gradient in units of K/km? A gradient of 25 K is huge and doesn't tell us anything about the stratification in the considered layer? Or do I misunderstand something here? Thanks for clarifying.
  - Since the vertical distance is approximately constant, the units are in principle interchangeable. Here we want to emphasize the importance of diabatic changes which are directly linked to changes of potential temperature. The difference in Theta between the two platforms is directly proportional to the Theta-gradient (more precise, the average of the gradient between the platforms). However, the vertical gradient of potential temperature is a common measure for static stability and carries more meaning than two values of potential temperature. The static stability above (25K/km is indeed huge!) as well as the neutral stability inside the cirrus are both consequences of the same process: diabatic cooling at the cloud top. The calculation of static stability is exactly the point of the measurement setup and this analysis.
- L 200: ...6.6 °° E  $\rightarrow$  6.6 °E Thanks for pointing that out! changed.
- L 200: One of the two mixing lines showing . . . . → which one? What does the other show?
   See next comment.
- L 200 207: I have difficulties to follow your description of Figure 4. Can you describe in more detail what exactly the two different mixing lines you are referring to are? You also say that RHi reaches 100% inside the cirrus layer? However, the values inside cirrus are already blue, so supersaturated? Thanks for clarifying. These changes address both comments, this and the one above.

Note that we slightly changed the layout of Fig. 4 (Fig. 2 in this document) according to the reviewers comments and updated the caption accordingly. We added clarifications to the description in the manuscript, which line corresponds to which section of the flight path:

The mixing line with lower N2O mixing ratios corresponds to the western part of the flight section at about 6.6°E, which is not influenced... ...However, on the mixing line with higher N2O mixing ratios, corresponding to the flight segment with cirrus occurrence at the Learjet level, relative humidity exceeds saturation (RHice  $\geq$  100 %) inside the cirrus and reaches up to RHice = 140 % in close vicinity to the cirrus in clear air.

- Fig. 5 d: x-axis, insert missing bracket fixed
- L 241: Over the analysed 2 h time window.... Shift this sentence to line 235, thus to description of figure 5. We agree, the sentence fits there better. changed
- Figure 6: missing bracket in x-axis description of panel b); figure caption last words: ...and no clouds (red) being present. In general, you could think about different colours for the red and green lines due to colour blindness.

Thank you for noticing the missing bracket and the comment regarding readability of plots. This has been fixed in the revised manuscript.

• L 242-249: you may want to shift the description of figure 5d to line 236 before you start the discussion of figure 6.

While we acknowledge that we jump back from Fig. 6 to Fig. 5 here, we think this sentence fulfills a purpose at this point in the text. Here we first mention the structure of humidity and thereby introduce the following paragraph. Therefore we would keep it that way.

- L 258: to investigate these processes we analyse... Thank you! changed.
- Figure 7: what are the vertical dashed lines? Why do the trajectories extend beyond the target region? Improve colour scale for panels a, d and g, because most of the values are in the green colour range.

  The plot segment to the right of the dashed line represents the time of measurement of the cirrus particles. We have also changed the range of the colourscale to make use of the full colour range. The trajectories extend beyond the measurement region as the simulations run slightly longer than 16 UTC.
- L 293: histograms in Fig. 8a
   Thanks for noticing! changed
- L 300-301: considering the parcel's initial PV .... What exactly do you mean here? Why is it important here and what does it tell us?

Fig. 8b uses the PV values in the measurement region for constructing the vertical axis. If instead the parcel's initial PV is used, the figure looks slightly different but overall similar (see. Fig. 4 in this reply document). This arises because PV change is not constant for all parcels and initial PV values. We choose to sort the parcels according to PV in the measurement area, because this is more consistent with other figures shown in the manuscript. It is important to not that the PV values used to sort the data point does not matter, because (i) PV is not necessarily a strictly vertically monotonically increasing property and (ii) diabatic PV change can result in vertical scrambling of air parcels, which would affect sedimentational and mixing redistribution of moisture.

We added some clarification to the manuscript:

This suggest that vertical scrambling of air parcels and non-uniformity of Lagrangian PV change does not affect our diagnostics of moisture change.

• Section 4.3: This section, including Figure 9 contains interesting and important information. However, the explanation in the text and the link to the figures are sometimes not easy to follow. So please make sure that all lines, hatching, dots, .... are explained in the figure caption of Figure 9 and refer to the plots in the text (including, white solid/dashed line, grey circles, dots, red hatched area, green solid line). For example, it would be very helpful to have the 2 pvu and 6 pvu isolines in each panel (are the white lines in panel b the pv isolines?). In panel b I would also improve the colour of the PV values. Now it is not easy to see the values precisely, one can only estimate the value at a given point.

Thank you for noting these details and unclear figure. We have added the missing information to the figure caption, added the 6 pvu isoline in all panels and made the colorscale in panel b more quantitative.

- L 360:  $8.7 \rightarrow 8.7$ Good catch!
- L 365: ... more tenuous cirrus further east (Fig. 10c). I have difficulties to see where the tenuous cirrus are, how they appear in panels c, f, i. Some more explanation would help.

This sentence refers to the northern part of the trajectory bundle. The bundle is marked in the figure. In general the difference signal between the  $10.8~\mu m$  and  $8.7~\mu m$  channel shows tenuous cirrus in white and other regions (with optically thick or no cirrus) in greyish colors. The difference channel is particularly designed to highlight optically thin cirrus and should be used in combination with the IR channels (e.g. shown in panels b, e, h) to obtain a full picture of cirrus distribution. The meaning of the channels was already explained in the original manuscript in L358-360, but we have slightly expanded this:

In the latter, optically thin cirrus clouds are shown in whitish colors. MSG images for all these channel (combinations) are shown for 01 UTC, 07 UTC and 13 UTC on 07 May 2013 in Fig. 10.

- L 389: You state that the moist layer is not as sharply capped in the model as in the observations. However, I cannot see this in the lines in Fig. 11b.
  - The dent in the humidity profile is visible in the line corresponding to the Stuttgart sounding (now blue, dark red in the previous version) at around 320K
- Figure 11: I would use the same colours for the triangles indicating the radiosonde stations and the lines in panel b). For example, the triangle for Oberschleissheim should be yellow. I would also either remove the WCB outflow, because this is not mentioned in the text describing Figure 11 or add the text mentioning the importance of the WCB outflow that is in the summary (L492) also here.

We have changed the color of the lines in Fig. 11b and have removed the WCB outflow from the figure. Also we have modified the text relating to WCB outflow in the summary.

We changed the text and caption accordingly.

• L420: ... a negative PV anomaly develops. Is it really a negative PV anomaly or is just the difference negative? In general, it is difficult to relate the text to the figure. Can you please improve this and point more precisely to the regions you are referring to.

It is a negative PV **anomaly** in the sense of difference between the two simulations. PV becomes not negative in either simulation, else we would have referred to a negative PV.

We have clarified this in the text.

- L429: ...., which is characterised by
  - That's how it was meant:
  - ... between troposphere and stratosphere. The two spheres, which are characterized by ...
- L459: I don't understand what you mean here with "making substantial STE during the transit unlikely". Where does this conclusion come from?

Good point! One sentence earlier we mentioned trajectories changing PV from < 2pvu to > 2pvu  $\rightarrow$  STE. This is however only a small fraction of trajectories and has no impact on the moisture nor does it explain the cirrus particles in the EXTL. The important point here is that the cirrus itself did not simply move from the troposphere into the stratosphere but formed there in situ while the necessary moisture was already there. We added clarification to the text:

There is no indication in the model, that the ExTL air mass gained additional moisture during this transit through exchange with the troposphere. Most importantly there is no indication of substantial STE of ice particles during the transit.